# Delayed Momentum Aggregation: Communication-efficient Byzantine-robust Federated Learning with Partial Participation

**Kaoru Otsuka** [1]  **Yuki Takezawa** [2]  **Makoto Yamada** [1]

## Abstract

Partial participation is essential for communication-efficient federated learning at scale, yet existing Byzantine-robust methods typically assume full client participation. In the partial participation setting, a majority of the sampled clients may be Byzantine, once Byzantine clients dominate, existing methods break down immediately. We introduce *delayed momentum aggregation*, a principle where the central server aggregates cached momentum from non-sampled clients along with fresh momentum from sampled clients. This principle ensures Byzantine clients remain a minority from the server's perspective even when they dominate the sampled set. We instantiate this principle in our optimizer DeMoA. We analyze the convergence rate of DeMoA, showing that DeMoA is Byzantine-robust under partial participation. Experiments show that, with $20\%$ Byzantine ratio and only $10\%$ partial participation rate, DeMoA achieves the best accuracy even when existing methods fail empirically.

## 1. Introduction

Federated Learning (FL) enables collaborative training across many clients without centralizing raw data, and has become a standard approach when privacy, bandwidth, or governance constraints prevent data pooling (Kairouz et al., 2021; McMahan et al., 2017). Its central idea is to transmit gradients rather than raw data. Specifically, each client computes gradients using their local dataset and sends it to the central server. Then, the central server computes the average of the gradients and updates the parameters. Since its proposal, FL has attracted many optimization researchers and has been widely studied in areas such as communication compression (Alistarh et al., 2017; Stich et al., 2018), data heterogeneity (Karimireddy et al., 2020; Wang et al., 2020), accelerated methods (Nesterov, 2018; Lin et al., 2015), and Byzantine-robust FL, including defenses for homogeneous data (Mhamdi et al., 2018; Blanchard et al., 2017; Pillutla et al., 2022; Karimireddy et al., 2021) and heterogeneous data (Xie et al., 2019b; Li et al., 2019; Karimireddy et al., 2022; Allouah et al., 2023a).

With many clients participating in training, FL systems are vulnerable to Byzantine clients, those that behave incorrectly due to faults or malicious intent (Kairouz et al., 2021; Lamport et al., 2019). Achieving Byzantine robustness requires two mechanisms. First, *robust aggregation*: naive averaging is vulnerable because even one Byzantine client can dominate the result; robust aggregators such as coordinate-wise median address this (Blanchard et al., 2017; Allouah et al., 2023a; Karimireddy et al., 2021). Second, *variance reduction*: momentum (Mhamdi et al., 2021; Karimireddy et al., 2021; Farhadkhani et al., 2022) or explicit variance reduction (Gorbunov et al., 2023; Rammal et al., 2024) helps distinguish Byzantine-induced drift from stochastic noise.

While these mechanisms are well-understood, their effectiveness hinges on a critical assumption: most existing Byzantine-robust FL methods require that all clients participate in every round, an assumption that is rarely met in practice. Under partial participation, this assumption breaks down, and as a consequence, most existing Byzantine-robust FL methods fail to remain robust against Byzantine clients. Specifically, under partial participation, some rounds may see the sampled clients form a Byzantine majority, which we refer to as a Byzantine majority round. In such a case, any robust aggregator that relies on sampled parameters no longer provides a good approximation of the average of the honest clients' parameters. Consequently, only a limited number of works have explicitly addressed Byzantine-robust FL under partial participation (Allouah et al., 2024; Malinovsky et al., 2024). However, these approaches either rely on mechanisms ineffective for deep learning settings (e.g., requiring large minibatches) or provide no mitigation against Byzantine majority rounds.

**Our Contributions.** In this paper, we tackle the challenge of Byzantine-robust FL with partial participation, aiming for

[1]Okinawa Institute of Science and Technology, Japan [2]Toyota Motor Corporation, Japan. Correspondence to: Kaoru Otsuka <kaoru.otsuka@oist.jp>.

*Proceedings of the $43^{rd}$ International Conference on Machine Learning*, Seoul, South Korea. PMLR 306, 2026. Copyright 2026 by the author(s).

a simple, practical, and provable solution. Our contributions are summarized as follows:

- We introduce the *delayed momentum aggregation* principle, where the server aggregates not only parameters from sampled clients but also the most recently received parameters from non-sampled clients. This ensures Byzantine clients never form a majority in any aggregation round.

- We instantiate this principle in DeMoA (Delayed Momentum Aggregation), a simple and practical algorithm with no additional communication overhead.

- We analyze the convergence rate of DeMoA under standard assumptions and show that DeMoA remains robust against Byzantine attacks, even with partial participation.

- Experiments on ResNet-18/CIFAR-10 confirm that DeMoA maintains stable training under partial participation where existing methods diverge.

## 2. Preliminary

**Notations.** Our notation largely follows (Karimireddy et al., 2022). We denote by $n$ the total number of clients, and for any positive integer $k$, let $[k] := \{1, 2, \ldots, k\}$. The set of honest (non-Byzantine) clients is represented by $\mathcal{G} \subseteq [n]$ with cardinality $G := |\mathcal{G}|$. The Byzantine ratio is defined as $\delta := (n - G)/n$, and throughout this paper we assume $\delta < 1/2$. For each client $i$, let $\mathcal{D}_i$ denote the distribution of local data $\xi_i$ over parameter space $\Omega_i$. The local loss functions $f_i : \mathbb{R}^d \to \mathbb{R}$ are defined as $f_i(\boldsymbol{x}) := \mathbb{E}_{\xi_i \sim \mathcal{D}_i}[f_i(\boldsymbol{x}; \xi_i)]$ where $f_i : \mathbb{R}^d \times \Omega_i \to \mathbb{R}$ is a sample loss.

**Assumptions.** We adopt several standard assumptions widely used in the analysis of federated learning (Koloskova et al., 2020; Bottou et al., 2018; Lan, 2020).

**Assumption 2.1** ($L$-smoothness and lower boundedness). The global objective $f$ is $L$-smooth if the gradient is $L$-Lipschitz i.e., there exists $L \geq 0$ such that

$$\|\nabla f(\boldsymbol{x}) - \nabla f(\boldsymbol{y})\| \leq L\|\boldsymbol{x} - \boldsymbol{y}\|, \ \boldsymbol{x}, \boldsymbol{y} \in \mathbb{R}^d.$$

We further assume that the global objective admits a minimum $f^* := \min_{\boldsymbol{x} \in \mathbb{R}^d} f(\boldsymbol{x})$, and denote the initial suboptimality by $F^0 := f(\boldsymbol{x}^0) - f^*$.

**Assumption 2.2** (Bounded variance and unbiasedness). There exists a constant $\sigma \geq 0$ such that the variance of the stochastic gradients is uniformly bounded:

$$\mathbb{E}_{\xi_i \sim \mathcal{D}_i}\left[\|\nabla f_i(\boldsymbol{x}, \xi_i) - \nabla f_i(\boldsymbol{x})\|^2\right] \leq \sigma^2, \ \boldsymbol{x} \in \mathbb{R}^d, \ i \in [n].$$

We also assume stochastic gradients are unbiased, i.e.,

$$\mathbb{E}_{\xi_i \sim \mathcal{D}_i}[\nabla f_i(\boldsymbol{x}, \xi_i)] = \nabla f_i(\boldsymbol{x}), \ \boldsymbol{x} \in \mathbb{R}^d.$$

**Assumption 2.3** (($\zeta, B$)-heterogeneity). There exists constants $\zeta, B \geq 0$ such that the average distance of local

gradients from the global gradient is bounded above as:

$$\frac{1}{G} \sum_{i \in \mathcal{G}} \|\nabla f_i(\boldsymbol{x}) - \nabla f(\boldsymbol{x})\|^2 \leq \zeta^2 + B^2 \|\nabla f(\boldsymbol{x})\|^2, \ \boldsymbol{x} \in \mathbb{R}^d.$$

**Problem Definition.** The objective function to minimize in this work is the global average of honest clients' local functions, i.e.,

$$\min_{\boldsymbol{x} \in \mathbb{R}^d} f(\boldsymbol{x}), \quad \text{where } f(\boldsymbol{x}) := \frac{1}{G} \sum_{i \in \mathcal{G}} f_i(\boldsymbol{x}),$$

where $\boldsymbol{x} \in \mathbb{R}^d$ denotes the model parameters and $\mathcal{D}_i$ represents the dataset distribution of client $i$. In general, $\mathcal{D}_i \neq \mathcal{D}_j$, reflecting data heterogeneity across clients.

### 2.1. Background: Byzantine-Robust Federated Learning — Why Local Momentum Is Crucial

The full participation setting serves as the theoretical foundation for Byzantine-robust FL, where the fundamental challenge is to design a mechanism that maintains convergence guarantees despite any behaviors by Byzantine clients. The case of full participation has been extensively studied in the literature (Karimireddy et al., 2022; Allouah et al., 2023a; Gorbunov et al., 2023). From here, we briefly summarize the insights on Byzantine-robust FL following these works.

In a Byzantine-robust FL setting, robustness is typically achieved by replacing the naive average with a robust aggregator (e.g., median). However, a robust aggregator alone is insufficient: Byzantine clients can inject small perturbations within the variance of honest gradients each round, which accumulate over time and cause divergence. This time-coupled attack, such as ALIE (Baruch et al., 2019), exploits the indistinguishability between stochastic noise and adversarial drift. To overcome this, variance reduction at each client (e.g., local momentum) is essential. By reducing stochastic noise at the client level, robust aggregators can reliably distinguish honest updates, providing theoretical guarantees against such attacks (Karimireddy et al., 2021).

The heterogeneous (non-IID) setting introduces additional difficulty, as Byzantine clients can exploit gradient heterogeneity. This necessitates pre-aggregation mechanisms such as bucketing (Karimireddy et al., 2022) or nearest-neighbor mixing (Allouah et al., 2023a), which average gradients within small groups before robust aggregation to reduce heterogeneity's impact.

While the precise definition of such aggregators may vary across works (Allouah et al., 2023a; Karimireddy et al., 2022; Gorbunov et al., 2023), we adopt the following definition from Gorbunov et al. (2023) throughout this paper.

**Definition 2.4** (($\delta, c$)-Robust Aggregator (Karimireddy et al., 2021; Gorbunov et al., 2023)). Assume that

$\{\boldsymbol{x}_1, \boldsymbol{x}_2, \ldots, \boldsymbol{x}_n\}$ is such that there exists a subset $\mathcal{G} \subseteq [n]$ of size $|\mathcal{G}| = G \geq (1 - \delta)n$ for $\delta < 1/2$, and there exists $\rho \geq 0$ such that $\frac{1}{G(G-1)} \sum_{i,j \in \mathcal{G}} \mathbb{E}\big[\|\boldsymbol{x}_i - \boldsymbol{x}_j\|^2\big] \leq \rho^2$, where the expectation is taken with respect to the randomness of $\{\boldsymbol{x}_i\}_{i \in \mathcal{G}}$. We say that $\hat{\boldsymbol{x}} = \mathrm{Agg}(\boldsymbol{x}_1, \ldots, \boldsymbol{x}_n)$ is a $(\delta, c)$-robust aggregator for some $c > 0$ if

$$\mathbb{E}\big[\|\hat{\boldsymbol{x}} - \bar{\boldsymbol{x}}\|^2\big] \leq c\delta\rho^2, \text{ where } \bar{\boldsymbol{x}} := \frac{1}{G} \sum_{i \in \mathcal{G}} \boldsymbol{x}_i.$$

Importantly, this definition is not merely abstract. Gorbunov et al. (2023) prove (Appendix D therein) that well-known aggregation rules such as `Krum` (Blanchard et al., 2017), `RFA` (Pillutla et al., 2022), and the coordinate-wise median (`CM`), when combined with *bucketing* technique (Karimireddy et al., 2022), indeed satisfy Definition 2.4. Thus, concrete, practical examples of robust aggregators are available within this framework.

## 2.2. Background: FL with Partial Participation

Federated learning with partial participation is a fundamental characteristic of practical federated learning systems. Real-world deployments inherently involve clients with heterogeneous capabilities and intermittent availability due to device constraints, battery limitations, and variations in network connectivity (McMahan et al., 2017; Kairouz et al., 2021). This participation pattern directly impacts communication efficiency and system scalability, making it a critical consideration for algorithm design.

In the usual partial participation setting, all clients are assumed to be non-Byzantine, i.e., $\delta = 0$. The classical algorithm `FedAvg` (McMahan et al., 2017) selects a subset of clients, denoted as $\mathcal{S}_t \subseteq [n]$, randomly at each round $t$. It then aggregates their local updates using the naive averaging i.e., $\frac{1}{|\mathcal{S}_t|} \sum_{i \in \mathcal{S}_t} \boldsymbol{g}_i^t$, where $\boldsymbol{g}_i^t$ represents the stochastic gradient of client $i$. If the stochastic gradient is replaced by the local momentum $\boldsymbol{m}_i^t$, the algorithm is referred to as `FedCM` (Xu et al., 2021).

## 2.3. Failure of Byzantine-robust Learning with Partial Participation

A naive extension of the full-participation Byzantine-robust learning result to partial participation combines robust aggregation with variance reduction (e.g., local momentum). It would replace naive averaging with robust aggregation over the sampled clients, i.e.,

$$\frac{1}{|\mathcal{S}_t|} \sum_{i \in \mathcal{S}_t} \boldsymbol{m}_i^t \quad \rightsquigarrow \quad \mathrm{Agg}(\{\boldsymbol{m}_i^t\}_{i \in \mathcal{S}_t}).$$

While appealing, **this strategy fails with partial participation**: in some rounds, the sampled set may contain a Byzantine majority, even when the (global) condition $\delta < 1/2$

holds. In classical settings where the server receives only the parameters sent in that round, no robust aggregator can reliably distinguish between Byzantine and honest updates when the sampled set contains a Byzantine majority. For example, if we consider uniform sampling with a fixed size $S$, the probability of at least one round containing a Byzantine-majority grows exponentially with iterations.

Two recent works take important steps toward addressing this challenge, but each comes with notable limitations. Allouah et al. (2024) provides the first characterizations of participation rates in Byzantine-robust FL; however, the required rates are too high for communication efficiency, the method does not address Byzantine majority rounds, and, without variance reduction, it fails under time-coupled attacks such as ALIE (Baruch et al., 2019). Malinovsky et al. (2024) successfully tolerate Byzantine majority rounds but require either large batch sizes or the computation of the full gradient, both of which are inefficient for deep learning (Defazio & Bottou, 2019).

## 3. Proposed Method

In this section, we propose *delayed momentum aggregation*. The main idea is to apply the robust aggregator not only to the momentum of sampled clients but also to the cached momentum of non-sampled clients. Then, we propose a *delayed momentum aggregation*-based optimizer `DeMoA`, which is Byzantine-robust even when the sampled clients form a Byzantine majority.

### 3.1. Main Idea: Delayed Momentum Aggregation.

Delayed momentum aggregation aggregates not only the momentum from the current round $t$, but also the most recent cached momentum maintained on the server, namely,

$$\mathrm{Agg}\left(\{\boldsymbol{m}_i^t\}_{i \in \mathcal{S}_t} \cup \{\mathcal{P}(\boldsymbol{m}_i^{t-\tau(i,t)}, i, t)\}_{i \in [n] \setminus \mathcal{S}_t}\right),$$

$$\text{(delayed momentum aggregation)}$$

where each $\boldsymbol{m}_i^t$ represents a local momentum, $\tau(i,t) := \min\{s \geq 0 : i \in \mathcal{S}_{t-s}\}$ denotes the delay since client $i$'s last update, and $\mathcal{P}$ is a (lightweight) preprocessing function that removes implicit momentum effect (Mitliagkas et al., 2016) from delayed momentum. The specific form of $\mathcal{P}$ depends on the underlying algorithm; we provide a concrete example in the following section. Delayed momentum aggregation maintains that $\mathrm{Agg}(\cdot)$ consistently sees the global Byzantine ratio $\delta < 1/2$, ensuring robustness even with partial participation.

**Intuition behind the principle** As established in prior works (Karimireddy et al., 2021; 2022), Byzantine robustness requires variance reduction and robust aggregators as core components. We introduce two key novelties:

First, we aggregate delayed momentum $\boldsymbol{m}_i^{t-\tau(i,t)}$ from non-sampled clients alongside fresh updates. With small stepsizes, the delayed momentum provides a good approximation of the fresh gradient $\nabla f_i(\boldsymbol{x}^t)$ as long as the delay $\tau(i,t)$ is not too large. This also helps reduce heterogeneity drift because the robust aggregator consistently receives gradient estimators from all honest clients, not just the sampled ones. While methods such as SCAFFOLD (Karimireddy et al., 2020) can further mitigate heterogeneity in non-Byzantine settings, they require sharing gradient correction terms that Byzantine clients could corrupt, making them unsuitable here. We demonstrate the effectiveness of this in Figure 2.

Second, the preprocessing function $\mathcal{P}$ removes additional variance introduced by the use of delayed momentum $\boldsymbol{m}_i^{t-\tau(i,t)}$. This design is motivated by prior work on asynchronous momentum SGD (Mitliagkas et al., 2016; Shi et al., 2024; Giladi et al., 2020).

We consider a specific instantiation of this principle in the next section; however, we believe this simple principle can be applied to other partial participation settings and optimizers as well, which we leave for future work.

### 3.2. Algorithm: DeMoA

As a concrete special case of the main idea, we propose a new optimizer, DeMoA, whose update rule is given in Algorithm 1. At each round $t$, the server independently samples each client with probability $p_t$ (i.e., $\boldsymbol{z}^t \sim \mathrm{Ber}(p_t)^{\otimes n}$ and $\mathcal{S}_t = \{i : z_i^t = 1\}$). The server refreshes the momentum of its sampled clients with the sampled ones, while updating the momentum of non-sampled clients using the preprocessing function $\mathcal{P}$ (which reweights the cached momentum here), namely,

$$
\boldsymbol{m}_i^t = \begin{cases} (1-\alpha_t p_t)\boldsymbol{m}_i^{t-1} + \alpha_t\,\nabla f_i(\boldsymbol{x}^{t-1},\xi_i^t), & i \in \mathcal{S}_t, \\ (1-\alpha_t p_t)\boldsymbol{m}_i^{t-1}, & i \notin \mathcal{S}_t, \end{cases}
$$

where $\alpha_t \in (0,1]$ is the local momentum parameter and each client $i$ is included in $\mathcal{S}_t$ with probability $p_t$.

**Why this (strange) design?** A key design choice is the use of $(1-\alpha_t p_t)$ as the momentum coefficient rather than the standard $(1-\alpha_t)$. While individual realizations of $\boldsymbol{m}_i^t$ are no longer convex combinations of the previous momentum and the gradient (i.e., $1-\alpha_t p_t + \alpha_t \neq 1$), this design choice ensures that explicit momentum $\alpha_t$ and implicit momentum effect induced by sampling probability $p_t$ are properly balanced in expectation. To see this, let $r_i^t \sim \mathrm{Ber}(p_t)$ denote whether client $i$ is sampled at round $t$. Then, we can rewrite the momentum update as

$$
\boldsymbol{m}_i^t = (1-\alpha_t p_t)\boldsymbol{m}_i^{t-1} + \alpha_t r_i^t\,\nabla f_i(\boldsymbol{x}^{t-1};\xi_i^t).
$$

Taking the conditional expectation over sampling random-

ness yields

$$
\mathbb{E}_{r_i^t}[\boldsymbol{m}_i^t \mid \boldsymbol{x}^{t-1}] = (1-\alpha_t p_t)\boldsymbol{m}_i^{t-1} + \alpha_t p_t\,\nabla f_i(\boldsymbol{x}^{t-1};\xi_i^t),
$$

recovering the standard momentum recursion in expectation with effective momentum parameter $\alpha_t p_t$.

The preprocessing function $\mathcal{P}$, which reweights delayed momentum by

$$
\mathcal{P}(\boldsymbol{m}_i^{t-\tau(i,t)},i,t) = \left[ \prod_{s=t-\tau(i,t)+1}^{t} (1-\alpha_s p_s) \right] \boldsymbol{m}_i^{t-\tau(i,t)},
$$

for a client with delay $\tau(i,t)$, removes additional variance introduced by delayed momentum $\boldsymbol{m}_i^{t-\tau(i,t)}$. To see why, consider an alternative design similar to MIFA (Gu et al., 2021) where sampled clients use momentum coefficient $(1-\alpha_t)$ and non-sampled clients keep them unchanged, i.e.,

$$
\boldsymbol{v}_i^t := \begin{cases} (1-\alpha_t)\boldsymbol{v}_i^{t-1} + \alpha_t\nabla f_i(\boldsymbol{x}^{t-1};\xi_i^t), & i \in \mathcal{S}_t, \\ \boldsymbol{v}_i^{t-1}, & i \notin \mathcal{S}_t. \end{cases}
$$

Both designs share the same conditional expectation given $\boldsymbol{x}^{t-1}$, but differ critically in variance. For simplicity, consider a deterministic gradient $\nabla f_i(\boldsymbol{x}^{t-1})$ instead of stochastic gradients. Then, our design satisfies

$$
\mathrm{Var}[\boldsymbol{m}_i^t \mid \boldsymbol{x}^{t-1}] = \alpha_t^2 p_t(1-p_t)\|\nabla f_i(\boldsymbol{x}^{t-1})\|^2,
$$

while the alternative incurs an additional term in its variance, namely, $\mathrm{Var}[\boldsymbol{v}_i^t \mid \boldsymbol{x}^{t-1}] = \alpha_t^2 p_t(1-p_t)\|\boldsymbol{v}_i^{t-1}\|^2 + \alpha_t^2 p_t(1-p_t)\|\nabla f_i(\boldsymbol{x}^{t-1})\|^2$. This extra variance arises because the momentum coefficient itself becomes random, i.e., $(1-\alpha_t r_i^t)\boldsymbol{v}_i^t + \alpha_t r_i^t\nabla f_i(\boldsymbol{x}^{t-1})$. By using $(1-\alpha_t p_t)$ uniformly, our design decouples these sources of randomness, keeping variance independent of the norm of delayed momentum $\|\boldsymbol{m}_i^{t-\tau(i,t)}\|$. A detailed derivation is provided in Appendix A.

**Communication efficiency.** DeMoA introduces no extra communication overhead. The server maintains a single vector $\boldsymbol{m}_i^t$ per client and reuses cached momentum (on server) for non-sampled clients, resulting in a memory requirement matching that of the full participation setting.

### 3.3. Theoretical Analysis

#### 3.3.1. MAIN RESULT

We analyze DeMoA under Assumptions 2.1, 2.2, and 2.3, with the $(\delta, c)$-robust aggregator (Definition 2.4), proving robustness to Byzantine clients even with partial participation (proof in Appendix E).

**Theorem 3.1.** *Suppose Assumptions 2.1, 2.2, and 2.3 hold. For a given $p \in (0,1]$, let the stepsize $\eta$, momentum $\alpha_t$, and*

**Algorithm 1** DeMoA: Delayed Momentum Aggregation-based SGD

**Require:** initial vectors $\boldsymbol{x}^0$, stepsize $\eta$, momentum parameter $\alpha_t$, robust aggregator Agg, client sampling probability $p_t \in (0, 1]$

1: Initialize $\boldsymbol{m}_i^0 = 0$ for all $i \in \mathcal{G}$.
2: **for** $t = 1, 2, \dots$ **do**
3:     Sample $\mathcal{S}_t \subseteq [n]$ by sampling each client $i \in [n]$ independently with probability $p_t$
4:     Server broadcasts $\boldsymbol{x}^{t-1}$ to all $i \in \mathcal{S}_t$
5:     **for all** $i \in \mathcal{S}_t$ (on sampled clients) **do**
6:         Draw $\xi_i^t \sim \mathcal{D}_i$ and compute

$$\boldsymbol{m}_i^t \leftarrow (1 - \alpha_t p_t)\boldsymbol{m}_i^{t-1} + \alpha_t \nabla f_i(\boldsymbol{x}^{t-1}; \xi_i^t)$$

7:         Send $\boldsymbol{m}_i^t$ to server
8:     **end for**
9:     **for all** $i \notin \mathcal{S}_t$ (on server) **do**
10:         $\boldsymbol{m}_i^t \leftarrow (1 - \alpha_t p_t)\boldsymbol{m}_i^{t-1}$
11:     **end for**
12:     $\boldsymbol{m}^t \leftarrow \mathrm{Agg}\Big(\{\boldsymbol{m}_i^t\}_{i \in \mathcal{S}_t} \cup \{\boldsymbol{m}_i^t\}_{i \notin \mathcal{S}_t}\Big)$
13:     $\boldsymbol{x}^t \leftarrow \boldsymbol{x}^{t-1} - \eta\,\boldsymbol{m}^t$
14: **end for**

*Bernoulli sampling probability $p_t$ be*

$$\eta := \min\Bigg\{ \frac{p}{\sqrt{90L}\max\{1, \Gamma\}},$$

$$\sqrt{\frac{8(f(\boldsymbol{x}^0) - f^*) + \frac{20\sigma^2}{\sqrt{90L}G} + \frac{24c\delta(2\sigma^2+\zeta^2)}{\sqrt{90L}}}{\Big[\Big(\frac{20L\sigma^2}{pG} + \frac{20L(1-p)\zeta^2}{pG}\Big) + \frac{60c\delta\sigma^2 L}{p}\Big]T}} \Bigg\},$$

$$\Gamma := \frac{(1-p)\left(20\left(1 + B^2\right) + 60c\delta G\right)}{G\left(1 - 60c\delta B^2\right)},$$

$p_t = p, \quad \alpha_t := \min(1, \sqrt{90L}\eta/p)$, *for all $t \geq 2$,*

*and $\alpha_1 = p_1 = 1$. Also, with condition on the Byzantine ratio $\delta$ as:*

$$\delta < \min\left(\frac{1}{2}, \frac{1}{60c(B^2 + \alpha(1-p))}\right).$$

*Then, the iterates $\{\boldsymbol{x}^t\}_{t=0}^{T-1}$ generated by the Algorithm 1 (DeMoA) satisfy*

$$\frac{1}{T}\sum_{t=0}^{T-1} \mathbb{E}\|\nabla f(\boldsymbol{x}^t)\|^2$$

$$\leq \mathcal{O}\Bigg( c\delta\zeta^2 + \sqrt{\frac{\left(LF^0 + c\delta(\sigma^2 + \zeta^2)\right)\left(\sigma^2 + (1-p)\zeta^2\right)}{pGT}}$$

$$+ \sqrt{\frac{c\delta\sigma^2\left(LF^0 + \frac{\sigma^2}{G} + c\delta(\sigma^2 + \zeta^2)\right)}{pT}} + \frac{\max\{1,\Gamma\}LF^0}{pT} \Bigg),$$

*where the expectation is taken over all sources of randomness in the algorithm.*

### 3.3.2. DISCUSSION

Theorem 3.1 shows that DeMoA maintains Byzantine robustness and converges to a neighborhood of stationarity in expectation for any sampling probability $p$, subject to the condition on $\delta$.

**Comparison to prior works.** Our result recovers existing convergence rates in special cases:

- **Full participation** ($p = 1, \delta \neq 0$): Our rate matches existing Byzantine-robust methods under full participation (Karimireddy et al., 2021; 2022; Allouah et al., 2023a). The dominant quadratic term, $\sqrt{LF^0 c\delta\sigma^2/(pT)}$, is optimal when $p = 1$ (Shi et al., 2025).
- **No Byzantine clients** ($\delta = 0, p \neq 1$): The sublinear component reduces to the standard partial participation rate, matching existing methods such as FEDCM (Xu et al., 2021).

**Non-vanishing term and significance of the principle.** Our guarantees ensure convergence to a $\mathcal{O}(c\delta\zeta^2)$ neighborhood of a stationary point in expectation. This non-vanishing term is unavoidable due to the combined effects of Byzantine clients and data heterogeneity, even in the full participation setting (Shi et al., 2025; Karimireddy et al., 2022). Furthermore, in sparse communication settings such as partial participation (Malinovsky et al., 2024; Allouah et al., 2024) or decentralized learning (He et al., 2022; Gaucher et al., 2025), existing methods suffer from an amplified non-vanishing term that depends on communication sparsity (e.g., $\mathcal{O}(c\delta\zeta^2/\gamma^2)$ for decentralized settings with a spectral gap $\gamma \leq 1$ depending on the communication topology). Our findings are significant because DeMoA prevents such amplification: the non-vanishing term remains $\mathcal{O}(c\delta\zeta^2)$ even under the sparse condition (i.e., partial participation). Moreover, as shown in Corollary 3.2, overparametrization can completely eliminate this non-vanishing term.

**Overparameterization fixes convergence.** In modern deep learning, models are overparameterized and exhibit *interpolation*-like behavior, so that gradients across clients tend to align as optimization progresses. A convenient way to capture this regime is the $(0, B)$-heterogeneity assumption (Karimireddy et al., 2022), which is also known as the *strong growth condition* in the optimization literature (Schmidt & Roux, 2013; Ma et al., 2018; Vaswani et al., 2019a;b; Meng et al., 2020). Under this assumption, we obtain the following corollary.

**Corollary 3.2** (Convergence under Overparameterization)**.** *Under the same conditions as in Theorem 3.1 with $(0, B)$-heterogeneity, the iterates $\{\boldsymbol{x}^t\}_{t=0}^{T-1}$ generated by the algo-*

*rithm 1 (*DeMoA*) satisfy*

$$\frac{1}{T}\sum_{t=0}^{T-1}\mathbb{E}\|\nabla f(\boldsymbol{x}^t)\|^2 = \mathcal{O}\left(\sigma\sqrt{\frac{(LF^0 + c\delta\sigma^2)}{pGT}}\right.$$

$$\left. + \sigma\sqrt{\frac{c\delta\left(LF^0 + \frac{\sigma^2}{G} + c\delta\sigma^2\right)}{pT}} + \frac{\max\{1,\Gamma\}LF^0}{pT}\right).$$

Compared to Theorem 3.1, the non-vanishing neighborhood term $\mathcal{O}(c\delta\zeta^2)$ disappears (since $\zeta = 0$), so we recover convergence to a stationary point in expectation. The remaining terms match the homogeneous ($\zeta = 0, B = 0$) rate up to constants, i.e., the same dependence on ($\sigma, p, G, T$) as in the optimal i.i.d. setting (Shi et al., 2025) when $p = 1$.

**Condition on $\delta$.** The condition on $\delta$, known as the breakdown point in robust statistics (Huber, 2005; Diakonikolas & Kane, 2023), is necessary for Byzantine-robust FL even with full gradient information. Allouah et al. (2023b) provides an upper bound $\delta < 1/(2 + B^2)$ for the full participation case ($p = 1$) under the ($\zeta, B$)-heterogeneity assumption. Our breakdown point matches this upper bound up to a constant factor. In partial participation ($p < 1$), let us think of the case when $\delta = \sigma = B = 0$, for simplicity. Then, $\alpha = 1$ and we have $\delta < \min(1/2, 1/(60c(1-p))$. Since typical aggregators have $c = \Theta(1)$ (Karimireddy et al., 2022; Gorbunov et al., 2023), $c = 10$ for example, then $\delta < 1/(600(1-p))$. This condition is mild because even in the extreme regime $p \to 0$, the restriction $\delta$ stays constant. We demonstrate that the restriction on $\delta$ does not significantly impact the optimization, based on experiments using various attacks under a low partial participation rate ($p = 0.1, \delta = 0.2$), as shown in Figure 1b. For additional experiments, see Appendix C.

**Robustness against Adaptive Attacks.** Our theoretical guarantee is derived under a worst-case adversary, so the convergence rate of DeMoA holds regardless of how Byzantine clients craft their updates, including adaptive strategies that target the delayed momentum principle. Empirically, the mimic attack (Karimireddy et al., 2022) in our experiments captures a natural adaptive strategy, but does not exhaust the space of attacks that could exploit the delayed momentum. We view dedicated attacks against the delayed momentum priciple, and corresponding empirical evaluation, as an important and valuable direction for future work.

## 4. Related Work

**Byzantine-robust FL with full participation.** Classical defenses replace naive averaging with robust aggregators: Krum (Blanchard et al., 2017), coordinate-wise median and trimmed-mean, geometric-median-based RFA (Pillutla et al., 2022), and meta-rules like Bulyan (Mhamdi et al., 2018). These aggregators alone are vulnerable to time-coupled attacks that inject small, indistinguishable biases accumulat-

ing across rounds (Baruch et al., 2019; Xie et al., 2019a). To counter this, Karimireddy et al. (2021); Mhamdi et al. (2021) show that variance reduction (e.g., local momentum) together with robust aggregation provably recovers convergence; subsequent work extends this view (Farhadkhani et al., 2022; Gorbunov et al., 2023). Under data heterogeneity, Byzantine clients can exploit heterogeneity to amplify drift even in the deterministic setting (Karimireddy et al., 2022). Pre-aggregation mechanisms such as bucketing (Karimireddy et al., 2022) and nearest-neighbor mixing (Allouah et al., 2023a) address this by averaging gradients within small groups before robust aggregation, closing the gap between achievable rates and lower bounds. Complementary directions include coding-theoretic redundancy (Chen et al., 2018), aggregation of double-momentum mechanism for SCO (Dahan & Levy, 2024a), near-optimal rates via aggregators from high-dimensional robust statistics (Zhu et al., 2023), and communication compression (Rammal et al., 2024; Gorbunov et al., 2023).

**Byzantine-robust FL with partial participation.** Partial participation introduces a fundamental challenge: even when the global Byzantine ratio satisfies $\delta < 1/2$, some rounds may sample a Byzantine majority. In such rounds, any aggregator relying solely on current inputs cannot reliably distinguish honest from adversarial updates. Early work assumes the sampled set always contains a sufficient honest fraction (Data & Diggavi, 2021), an assumption violated with high probability over many rounds. Malinovsky et al. (2024) tolerate Byzantine majority rounds via variance reduction with specialized clipping, but their method relies on large minibatches impractical for deep learning (Defazio & Bottou, 2019). Allouah et al. (2024) replace naive averaging in FedAvg with robust aggregation, yet use vanilla SGD, which is vulnerable to time-coupled attacks (Baruch et al., 2019; Karimireddy et al., 2021), and provide no mitigation when Byzantines dominate a round. Our work addresses both limitations: we incorporate momentum for time-coupled robustness and leverage historical information to ensure an honest majority at every aggregation.

**Use of Delayed Updates and asynchronous methods.** Our work is closely related to MIFA (Gu et al., 2021) and other similar methods such as Fedvarp (Jhunjhunwala et al., 2022), and CA²FL (Wang et al., 2024). All of the above methods address client unavailability by caching each client's latest update and substituting the global average with it when the client is absent, though their motivation is not for Byzantine robustness. Our method recovers MIFA by setting the momentum parameter $\alpha = 1$, the aggregator $\text{Agg}$ to naive averaging, and the preprocessing function to $\mathcal{P}(x, i, t) := x$; however, this yields vacuous bounds in the Byzantine setting since naive averaging corresponds to $c = \infty$. Beyond this, we differ from all the above methods in two key directions. First, the above methods only treat only SGD (at

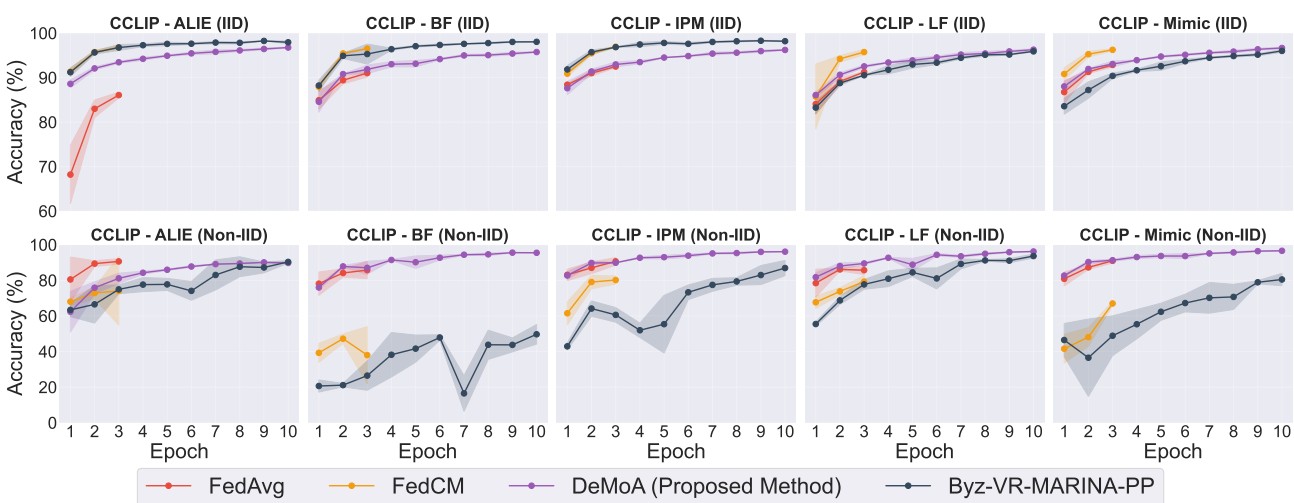

*(a)* ConvNet on MNIST with partial participation rate $p = 0.5$. With a high participation rate, Byzantine majority occurs later: FedAvg and FedCM trajectories are reported up to epoch 3, when the first Byzantine majority round occurs.

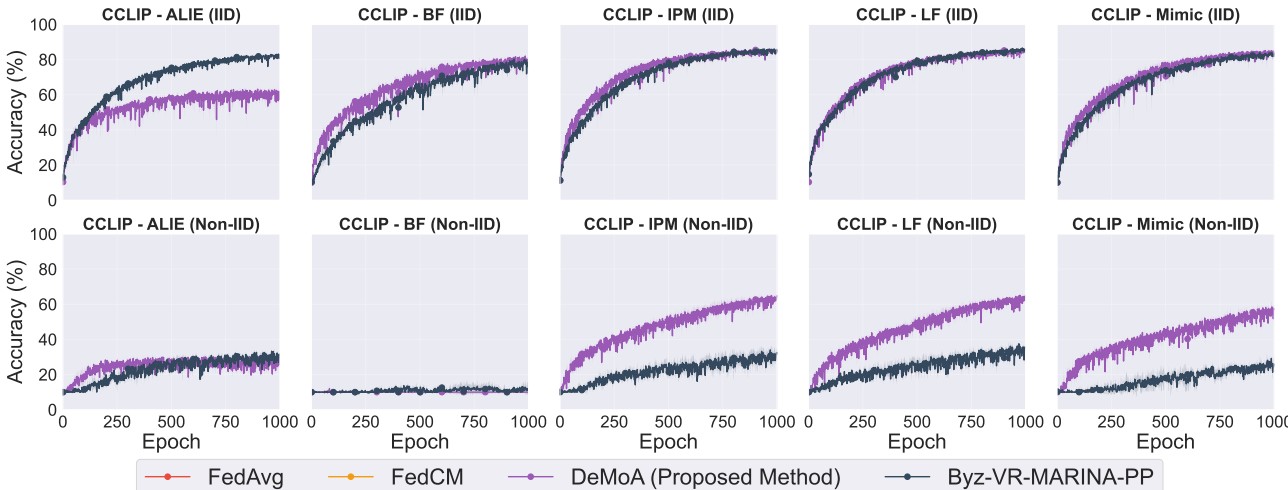

*(b)* ResNet-18 on CIFAR-10 with partial participation rate $p = 0.1$. With such low participation, FedAvg and FedCM face a **Byzantine majority round in the very first epoch (!)**, leading to immediate collapse.

*Figure 1.* Byzantine-robust training with Byzantine ratio $\delta = 0.2$ using centered clipping (CCLIP). Top: IID data; bottom: non-IID data; columns correspond to different attacks. FedAvg and FedCM eventually encounter a Byzantine majority round and collapse, while Byz-VR-MARINA-PP remains stable but attains lower test accuracy than DeMoA due to their bias from clipping.

least in their convergence analysis); even when paired with a robust aggregator, it remains vulnerable to time-coupled attacks, as discussed in the full participation setting. Second, naively changing SGD to momentum SGD introduces an implicit momentum effect from cached updates, which degrades convergence (Mitliagkas et al., 2016). We provide a perspective on this in Appendix A. Briefly, the use of delayed momentum along with partial participation introduces a variance that could lead to inefficiency for Federated Learning. Thus, careful correction for this implicit momentum effect is necessary when using delayed momentum. Asynchronous optimization literature (Mishchenko et al., 2022; Koloskova et al., 2022) often uses delayed gradients/momentum; OrMo (Shi et al., 2024) analyzes

asynchronous momentum SGD and proposes a correction mechanism for delayed momentum under the bounded gradient assumption, which is rarely met in practice. Our design of the preprocessing function is inspired by OrMo, though we operate in the partial participation setting rather than asynchronous setting. Additionally, Byzantine-robust asynchronous FL (Dahan & Levy, 2024b; Yang & Li, 2023; Xie et al., 2020; Fang et al., 2022) also addresses heterogeneous client delays, whereas we focus on partial participation in a synchronous setting.

**Communication Compression.** Approaches to communication efficiency in FL broadly fall into two categories. The first sparsifies communication via partial participation,

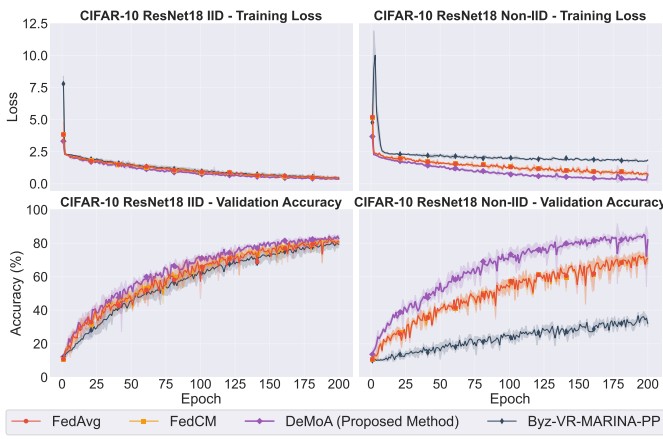

*Figure 2.* Training ResNet-18 on CIFAR-10 without Byzantine clients ($\delta = 0$) under partial participation rate $p = 0.5$ using naive averaging (avg). Top: training loss (lower is better); bottom: test accuracy (higher is better). Left: IID; right: non-IID. FedCM follows similar trajectories as FedAvg because the gains from explicit momentum vanish: the *implicit momentum effect* largely dominates the updates. In contrast, DeMoA mitigates this effect via *delayed momentum aggregation*, which remains effective even without Byzantine clients.

which is the setting studied in this paper and discussed in Section 2. The second compresses the transmitted updates themselves (e.g., gradients or momentum), a direction that has attracted substantial attention in the optimization literature (Alistarh et al., 2017; Khirirat et al., 2018; Stich et al., 2018). In the Byzantine-robust setting, Gorbunov et al. (2023); Rammal et al. (2024) study compression under full participation, while Malinovsky et al. (2024) considers compression under partial participation. These approaches are all MARINA-based, which makes their algorithmic structure naturally amenable to compression. However, the momentum-based extension of Byz-VR-MARINA-PP (Malinovsky et al., 2024) comes without a theoretical guarantee. Additionally, although their MARINA-based optimizer is provably convergent, variance-reduction based methods are known to perform poorly in deep learning (Defazio & Bottou, 2019), limiting its practical utility. DeMoA stores per-client momentum on the server, which can make memory a concern at scale. Compression can be naturally folded into the preprocessing function $\mathcal{P}$, and we leave a thorough treatment to future work.

# 5. Experiments

We evaluated DeMoA under various Byzantine attacks with partial participation by training a convolutional network on MNIST and a ResNet-18 on CIFAR-10 across IID and non-IID data distributions. We compared four optimizers (FedAvg, FedCM, DeMoA, and the heuristic momentum extension of Byz-VR-MARINA-PP (proposed by Malinovsky et al. (2024) for deep learning tasks) with four aggregators (Coordinate-wise Median CM, Krum (Blanchard et al.,

2017), RFA (Pillutla et al., 2022), CCLIP (Karimireddy et al., 2021)) under five Byzantine attacks (ALIE (Baruch et al., 2019), Bit-Flipping, IPM (Xie et al., 2019a), Label-Flipping, Mimic (Karimireddy et al., 2022)). FedAvg (McMahan et al., 2017) performed single-step SGD per client followed by server-side aggregation, while FedAvgM (Xu et al., 2021) with local (client-side) momentum ($\alpha = 0.9$). In our setting, the (naive) averaging step in the four optimizers was replaced by the robust aggregators, allowing us to assess performance under Byzantine attacks. Our implementation extended the codebase [1] with additional features from the ByzFL framework (González et al., 2025) and additional support for CIFAR-10/ResNet-18 training and partial participation. Appendix B provides complete experimental details.

**Hyperparameter selection.** For each optimizer (FedAvg, FedCM, DeMoA) we tuned a global learning rate $\eta$ over the grid $\{0.1, 0.01, 0.001\}$. Byz-VR-MARINA-PP required tuning both $\eta$ and the clipping radius $\lambda \in \{100.0, 10.0, 1.0\}$. Every configuration was evaluated over seeds $\{0, 1, 2\}$, and we selected the setting with the highest mean validation accuracy for reporting in both the non-Byzantine and Byzantine settings. The shaded regions in all figures reflect the standard deviation across seeds for each chosen hyperparameter configuration.

## 5.1. Byzantine Robustness with Partial Participation

We studied the setting of partial participation rates $p = 0.5$ and $p = 0.1$ along with $n = 25$ total clients, among which 20% were Byzantine ($\delta = 0.2$), on both settings. Figures 1a and 1b illustrate the performance of the optimizers using the CCLIP aggregator under Byzantine attacks with partial participation. Our experiments lead to three key observations.

DeMoA **consistently achieves the highest final accuracy across most settings.** On both CIFAR-10 and MNIST, DeMoA attains higher and more stable accuracy across all attacks. In contrast, FedAvg and FedCM collapse early during training: in the CIFAR-10 experiments, they fail in the very first epoch (Fig. 3), while in the MNIST experiments, they collapse by epoch 4.

**Non-IID data highlight accuracy differences.** Under non-IID data distributions on both CIFAR-10 and MNIST (lower halves of Figs. 1a and 1b), Byz-VR-MARINA-PP exhibits high variance, as indicated by the shaded regions, and unstable convergence, sometimes failing catastrophically. In contrast, DeMoA remains robust and converges reliably across most settings.

**Performance varies across aggregators.** Similar qualitative trends are observed for other aggregators (krum,

---

[1] https://github.com/epfml/byzantine-robust-noniid-optimizer

cm, and `rfa`) and across both datasets. However, for certain aggregators, DeMoA performs poorly. This behavior arises when the combination of partial participation rate and Byzantine ratio exceeds the breakdown point of the corresponding aggregator. The complete set of results is reported in Appendix C.

## 5.2. Baseline Experiments without Byzantine Clients

We also examined the non-Byzantine setting ($\delta = 0$) to establish baseline performance. The setup used $n = 20$ clients with the `avg` aggregator. The results are summarized in Figure 2.

**DeMoA beats standard methods even without Byzantine clients.** Surprisingly, DeMoA consistently outperformed FedCM in the non-Byzantine setting ($\delta = 0$), despite the risk that reusing momentum across rounds could degrade performance. The curves suggested that with partial participation and heterogeneity, the delayed momentum aggregation principle in DeMoA mitigated drifts induced by heterogeneity, acting as an *implicit regularizer* even without attacks. Across both IID and non-IID settings, Byz-VR-MARINA-PP achieved the worst validation accuracy and highest loss throughout training. This performance degradation stems from applying clipping to momentum differences, which introduces a bias that harms performance unless the clipping hyperparameter $\lambda$ is selected with particular care, i.e., an additional grid search over $\lambda \in \{100.0, 10.0, 1.0\}$. This highlights an inherent trade-off: although clipping is essential for defending against Byzantine behaviors, clipping can substantially worsen gradient estimates.

## 6. Conclusion

We introduced the *delayed momentum aggregation* principle, in which the server aggregates parameters from sampled clients along with the most recently received parameters from non-sampled clients, ensuring that Byzantine clients never form a majority in any aggregation round. We instantiated this principle in DeMoA, a simple algorithm with no additional communication overhead, and provided convergence guarantees matching the fundamental lower bounds for full participation. Under overparametrization, the non-vanishing error caused by Byzantine clients and heterogeneity disappears entirely. Experiments on ResNet-18/CIFAR-10 demonstrated that DeMoA maintains stable training even with partial participation, whereas existing methods diverge under Byzantine attacks. The delayed momentum aggregation principle opens promising avenues for extension to other client selection schemes (Fraboni et al., 2022; Cho et al., 2020; Fraboni et al., 2021; Li et al., 2020; Chen et al., 2022) beyond Bernoulli sampling.

## Impact Statement

This paper presents work whose goal is to advance the field of federated learning by improving its robustness against adversarial participants. Federated learning enables collaborative model training while keeping data decentralized, which benefits privacy-sensitive applications in healthcare, finance, and mobile computing. Our proposed method is purely defensive: it enhances the reliability and safety of distributed training systems by preventing malicious participants from corrupting the learned model. We use the term "Byzantine" following established distributed systems nomenclature to denote arbitrary failures, with no cultural reference intended. We do not foresee negative societal consequences specific to this work beyond those common to general advances in machine learning.

## Acknowledgement

Makoto Yamada was partly supported by JSPS KAK-ENHI Grant Number 24K03004 and by JST ASPIRE JP-MJAP2302. Yuki Takezawa was supported by JSPS KAK-ENHI Grant Number 23KJ1336. This work was conducted while Yuki Takezawa was affiliated with Kyoto University and OIST.

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

# A. Algorithm Details

In this appendix, we provide the detailed derivation showing how DeMoA instantiates the delayed momentum aggregation principle introduced in Section 1. We begin by establishing the connection between the general principle and our specific algorithm, then explain the design choices that ensure both Byzantine robustness and optimal convergence.

## A.1. Instantiation of Delayed Momentum Aggregation

Recall that the **(delayed momentum aggregation)** principle updates the global model as:

$$\boldsymbol{x}^t = \boldsymbol{x}^{t-1} - \eta \, \mathrm{Agg} \left( \{\boldsymbol{m}_i^t\}_{i \in \mathcal{S}_t} \cup \{\mathcal{P}(\boldsymbol{m}_i^{t-\tau(i,t)}, i, t)\}_{i \in [n] \setminus \mathcal{S}_t} \right)$$

where $\tau(i, t)$ denotes the delay since client $i$'s last update and $\mathcal{P}$ is a preprocessing function that removes implicit momentum bias (Mitliagkas et al., 2016).

To see how DeMoA implements this principle, define $\tau(i, t) = \min\{s \geq 0 : i \in \mathcal{S}_{t-s}\}$ as the number of rounds since client $i$ last participated. When $i \in \mathcal{S}_t$, we have $\tau(i, t) = 0$ (fresh update); when $i \notin \mathcal{S}_t$, we have $\tau(i, t) > 0$ (stale update).

For the moment, let us pick $\alpha_t = \alpha, p_t = p$ for simplicity, then the preprocessing function in DeMoA is instantiated as:

$$\mathcal{P}(\boldsymbol{m}_i^{t-\tau(i,t)}, i, t) := (1 - \alpha p)^{\tau(i,t)} \boldsymbol{m}_i^{t-\tau(i,t)}$$

This corresponds exactly to the server-side update for non-sampled clients (Algorithm 1, line 10): for each round that client $i$ is not sampled, its cached momentum is multiplied by $(1 - \alpha p)$. Thus, after $\tau(i, t)$ rounds without participation, the cached momentum becomes $(1 - \alpha p)^{\tau(i,t)} \boldsymbol{m}_i^{t-\tau(i,t)}$, which is precisely what the server maintains as $\boldsymbol{m}_i^t$ in the algorithm.

This ensures that the aggregator consistently operates on all $n$ clients (both sampled and non-sampled), maintaining the global Byzantine ratio $\delta < 1/2$ even when the sampled set contains a Byzantine majority.

## A.2. Design Choice: Explicit Momentum Correction

The update rule for sampled clients in DeMoA uses the momentum coefficient $(1 - \alpha p)$ rather than the standard $(1 - \alpha)$. We now explain why this design is crucial.

### A.2.1. THE PROPOSED DESIGN

In each round $t$, the server independently samples each client with probability $p$ (i.e., $z^t \sim \mathrm{Ber}(p)^{\otimes n}$ and $\mathcal{S}_t = \{i : z_i^t = 1\}$). The selected clients refresh their momentum using:

$$\boldsymbol{m}_i^t := \begin{cases} (1 - \alpha p)\boldsymbol{m}_i^{t-1} + \alpha \, \nabla f_i(\boldsymbol{x}^{t-1}, \xi_i^t), & i \in \mathcal{S}_t, \\ (1 - \alpha p)\boldsymbol{m}_i^{t-1}, & i \notin \mathcal{S}_t, \end{cases}$$

where $\alpha \in (0, 1]$ is the client momentum parameter.

Taking the conditional expectation over the sampling randomness at round $t$ given the filtration $\mathcal{F}_{t-1}$, we obtain

$$\mathbb{E}[\boldsymbol{m}_i^t \mid \mathcal{F}_{t-1}] = (1 - \alpha p)\boldsymbol{m}_i^{t-1} + \alpha p \, \nabla f_i(\boldsymbol{x}^{t-1}).$$

This shows that the explicit momentum coefficient $\alpha$ and the implicit momentum induced by sampling probability $p$ are properly balanced.

### A.2.2. AN ALTERNATIVE DESIGN AND ITS SUBOPTIMALITY

One might consider an alternative design (e.g., naively incorporating MIFA's principle (Gu et al., 2021) where sampled clients use only the explicit momentum:

$$\boldsymbol{v}_i^t := \begin{cases} (1 - \alpha)\boldsymbol{v}_i^{t-1} + \alpha \, \nabla f_i(\boldsymbol{x}^{t-1}, \xi_i^{t-1}), & i \in \mathcal{S}_t, \\ \boldsymbol{v}_i^{t-1}, & i \notin \mathcal{S}_t. \end{cases}$$

This alternative has the same conditional expectation:

$$\mathbb{E}[\boldsymbol{v}_i^t \mid \mathcal{F}_{t-1}] = (1 - \alpha p)\boldsymbol{v}_i^{t-1} + \alpha p \, \nabla f_i(\boldsymbol{x}^{t-1}).$$

However, this design is suboptimal due to increased variance caused by the implicit momentum effect from client sampling. Let $r_i^t \sim \mathrm{Ber}(p)$ denote the indicator of whether client $i$ is sampled at round $t$ (i.e., $r_i^t = 1$ if client $i$ is sampled and $r_i^t = 0$ otherwise). The conditional variance of our proposed design satisfies

$$
\begin{aligned}
\mathrm{Var}[\boldsymbol{m}_i^t \mid \mathcal{F}_{t-1}] &:= \mathbb{E}\left[\left\|(1-\alpha p)\boldsymbol{m}_i^{t-1} + \alpha r_i^t \, \nabla f_i(\boldsymbol{x}^{t-1}, \xi_i^t) - \left((1-\alpha p)\boldsymbol{m}_i^{t-1} + \alpha p \, \nabla f_i(\boldsymbol{x}^{t-1})\right)\right\|^2 \mid \mathcal{F}_{t-1}\right] \\
&= \mathbb{E}\left[\|\alpha r_i^t(\nabla f_i(\boldsymbol{x}^{t-1}; \xi_i^t) - \nabla f_i(\boldsymbol{x}^{t-1})) + \alpha(r_i^t - p)\nabla f_i(\boldsymbol{x}^{t-1})\|^2 \mid \mathcal{F}_{t-1}\right] \\
&= \alpha^2\mathbb{E}[|r_i^t|^2 \mid \mathcal{F}_{t-1}]\mathbb{E}\left[\|\nabla f_i(\boldsymbol{x}^{t-1}; \xi_i^t) - \nabla f_i(\boldsymbol{x}^{t-1})\|^2 \mid \mathcal{F}_{t-1}\right] + \alpha^2\mathbb{E}[|r_i^t - p|^2 \mid \mathcal{F}_{t-1}]\|\nabla f_i(\boldsymbol{x}^{t-1})\|^2 \\
&= \alpha^2 p\, \mathbb{E}\left[\|\nabla f_i(\boldsymbol{x}^{t-1}; \xi_i^t) - \nabla f_i(\boldsymbol{x}^{t-1})\|^2 \mid \mathcal{F}_{t-1}\right] + \alpha^2 p(1-p)\|\nabla f_i(\boldsymbol{x}^{t-1})\|^2 \\
&\le p\alpha^2\sigma^2 + \alpha^2 p(1-p)\|\nabla f_i(\boldsymbol{x}^{t-1})\|^2,
\end{aligned}
$$

where the third equality follows from the unbiasedness and the independence between the stochastic gradient and the Bernoulli random variable $r_i^t$, the fourth equality follows from standard properties of Bernoulli random variables, and the final inequality follows from Assumption 2.2.

While the alternative design has

$$
\begin{aligned}
\mathrm{Var}[\boldsymbol{v}_i^t \mid \mathcal{F}_{t-1}] &:= \mathbb{E}\left[\left\|(1-\alpha r_i^t)\boldsymbol{v}_i^{t-1} + \alpha r_i^t \, \nabla f_i(\boldsymbol{x}^{t-1}, \xi_i^t) - \left((1-\alpha p)\boldsymbol{v}_i^{t-1} + \alpha p \, \nabla f_i(\boldsymbol{x}^{t-1})\right)\right\|^2 \mid \mathcal{F}_{t-1}\right] \\
&= \mathbb{E}\left[\left\|\alpha(p - r_i^t)\boldsymbol{v}_i^{t-1} + \alpha r_i^t(\nabla f_i(\boldsymbol{x}^{t-1}; \xi_i^t) - \nabla f_i(\boldsymbol{x}^{t-1})) + \alpha(r_i^t - p)\nabla f_i(\boldsymbol{x}^{t-1})\right\|^2 \mid \mathcal{F}_{t-1}\right] \\
&\le \alpha^2 p(1-p)\|\boldsymbol{v}_i^{t-1}\|^2 + p\alpha^2\sigma^2 + \alpha^2 p(1-p)\|\nabla f_i(\boldsymbol{x}^{t-1})\|^2.
\end{aligned}
$$

where the inequalities follow from Assumption 2.2.

The additional variance term $\alpha^2 p(1 - p)\|\boldsymbol{v}_i^{t-1}\|^2$ arises from the implicit momentum effect: when client sampling is stochastic, the momentum coefficient itself becomes random in the alternative design, introducing extra variance. This additional variance propagates through the convergence analysis.

### A.3. Communication and Memory Efficiency

Importantly, DeMoA does not incur additional communication costs compared to standard partial participation methods. In each round, the server only communicates with the sampled clients in $\mathcal{S}_t$, maintaining the communication efficiency benefits of partial participation. The server stores one momentum vector $\boldsymbol{m}_i^t$ per client, matching the memory requirements of full participation Byzantine-robust methods while supporting arbitrary partial participation patterns.

## B. Additional Experimental Details

### B.1. Common Experimental Settings

All experiments covered two standard computer vision tasks: MNIST with a convolutional neural network architecture (CONV-CONV-DROPOUT-FC-DROPOUT-FC) and CIFAR-10 with a standard ResNet-18. Training employed cross-entropy (negative log-likelihood) loss with batch size 32 per client and partial participation rate $p \in \{0.1, 0.5\}$. We evaluated both IID and non-IID data partitions, with the latter following the class-based approach of Karimireddy et al. (2022). Four optimizers were compared: FedAvg, FedCM, DeMoA, and the heuristic momentum extension of Byz-VR-MARINA-PP introduced in (Malinovsky et al., 2024), all using momentum parameter $\alpha = 0.9$ where applicable. Training ran for 10 epochs (300 iterations total) for MNIST and 1000 epochs for CIFAR-10, with results averaged over seeds $\{0, 1, 2\}$. For each optimizer we tuned the learning rate $\eta \in \{0.1, 0.01, 0.001\}$; additionally Byz-VR-MARINA-PP jointly tuned the clipping radius $\lambda \in \{100.0, 10.0, 1.0\}$. We selected the configuration with the highest mean validation accuracy across the three seeds for both the non-Byzantine and Byzantine experiments. Tables 1, 2 and 3 provided complete configuration details. We also conducted experiments under additional settings of participation rate $p \in \{0.1, 0.5\}$ and momentum parameter $\alpha = 0.9$; these extended results are presented in Appendix C.

*Table 1.* Byzantine MNIST training configuration used in Fig. 1a, 4a, 4b, 5a, 5b, 6a, 6b.

| | |
|---|---|
| Dataset | MNIST (IID and non-IID with bucketing $s = 2$) |
| Model | CONV-CONV-DROPOUT-FC-DROPOUT-FC |
| Clients | $n = 25$ (20 honest, 5 Byzantine; $\delta = 0.2$) |
| Participation | Both experiments on $p = 0.1, 0.5$ |
| Momentum | $\alpha = 0.9$ |
| Aggregators | `krum`, `cm`, `CCLIP`, `rfa` |
| Batch size | 32 per client |
| Training horizon | 10 epochs (300 rounds) |
| Attacks | BF, LF, mimic, IPM, ALIE |
| Optimizers | FedAvg, FedCM, DeMoA, Byz-VR-MARINA-PP |
| Learning-rate tuning | grid search on $\{0.1, 0.01, 0.001\}$ |
| Byz-VR-MARINA-PP tuning | joint grid search $\lambda \in \{100.0, 10.0, 1.0\}$ |
| Seeds | $\{0, 1, 2\}$ |

*Notation:* `avg`=naive average, `krum`=Krum (Blanchard et al., 2017), `cm`=coordinate-wise median, `CCLIP`=centered clipping (Karimireddy et al., 2021), `rfa`=geometric median (RFA) (Pillutla et al., 2022).

*Table 2.* CIFAR-10 (non-Byzantine) configuration used in Fig. 2.

| | |
|---|---|
| Dataset | CIFAR-10 (IID and non-IID partitions with bucketing $s = 2$) |
| Model | ResNet-18 |
| Clients | $n = 20$ (all honest) |
| Participation | $p = 0.5$ |
| Momentum | $\alpha = 0.9$ |
| Aggregator | naive averaging `avg` |
| Batch size | 32 per client |
| Training horizon | 200 epochs |
| Optimizers | FedAvg, FedCM, DeMoA, Byz-VR-MARINA-PP |
| Learning-rate tuning | grid search on $\{0.1, 0.01, 0.001\}$ |
| Byz-VR-MARINA-PP tuning | joint grid search $\lambda \in \{100.0, 10.0, 1.0\}$ |
| Seeds | $\{0, 1, 2\}$ |

## B.2. Baseline Performance Evaluation

This experiment established baseline performance under partial participation without Byzantine clients across both MNIST (ConvNet) and CIFAR-10 (ResNet-18). We used $n = 20$ clients with no Byzantine clients ($\delta = 0$) and naive averaging as aggregation. The objective was to validate that DeMoA maintains performance even in non-Byzantine settings and to establish reference performance levels for subsequent robustness comparisons. A result in Fig. 2 demonstrated that DeMoA outperformed standard momentum methods on both MNIST and CIFAR-10 even without Byzantine clients, suggesting that delayed momentum aggregation provided *implicit regularization* benefits especially under heterogeneous data distributions.

## B.3. Byzantine Robustness Assessment

This experiment evaluated robustness against Byzantine attacks under partial participation on both datasets (MNIST with the ConvNet backbone and CIFAR-10 with ResNet-18). We configured $n = 25$ clients with 5 Byzantine clients (20%) for MNIST and $n = 25$ with 5 Byzantine clients (20%) for CIFAR-10. Four robust aggregators were evaluated: Krum, CM (coordinate-wise median), CCLIP (centered clipping), and RFA. The experimental design included both IID and non-IID data partitions, with bucketing applied in the Byzantine non-IID setting to mitigate extreme heterogeneity with bucketing size $s = 2$.

## B.4. Non-IID data partition

We constructed the non-IID split following Karimireddy et al. (2022) in the *balanced* case: (i) sorted the training sets by label; (ii) split it into $G$ equal, contiguous shards (where $G$ is the number of honest clients); (iii) assigned one shard to each honest client and shuffle examples within each client. We partitioned the test set analogously.

*Table 3.* CIFAR-10 (Byzantine) configuration used in Fig. 1b .

| | |
|---|---|
| Dataset | CIFAR-10 (IID and non-IID with bucketing $s = 2$) |
| Model | ResNet-18 |
| Clients | $n = 25$ (20 honest, 5 Byzantine; $\delta = 0.2$) |
| Participation | $p = 0.1$ |
| Aggregators | CCLIP |
| Batch size | 32 per client |
| Training horizon | 1000 epochs |
| Attacks | BF, LF, mimic, IPM, ALIE |
| Optimizers | FedAvg, FedCM, DeMoA, Byz-VR-MARINA-PP |
| Learning-rate tuning | grid search on $\{0.1, 0.01, 0.001\}$ |
| Byz-VR-MARINA-PP tuning | joint grid search $\lambda \in \{100.0, 10.0, 1.0\}$ |
| Seeds | $\{0, 1, 2\}$ |

*Notation:* krum=Krum (Blanchard et al., 2017), cm=coordinate-wise median, CCLIP=centered clipping (Karimireddy et al., 2021), rfa=geometric median (RFA) (Pillutla et al., 2022).

## B.5. Definition of Robust Aggregators Used

Let $\boldsymbol{x}_1, \ldots, \boldsymbol{x}_n \in \mathbb{R}^d$ denote the inputs to an aggregator (in experiments, these correspond to the per-client updates submitted to the server in a given round), and let $\delta \in [0, 1/2)$ denote the Byzantine fraction. We use $[\boldsymbol{x}]_k$ to denote the $k$-th coordinate of a vector $\boldsymbol{x}$.

**Coordinate-wise Median (CM).** Aggregate by taking the median in each coordinate:

$$\big[ \mathrm{CM}(\boldsymbol{x}_1, \ldots, \boldsymbol{x}_n) \big]_k = \mathrm{median}\big([\boldsymbol{x}_1]_k, \ldots, [\boldsymbol{x}_n]_k\big), \qquad k = 1, \ldots, d.$$

**Krum (KRUM).** Krum selects the input that is closest to its $(n - \delta n - 2)$ nearest neighbors in $\ell_2$ distance:

$$\mathrm{Krum}(\boldsymbol{x}_1, \ldots, \boldsymbol{x}_n) = \arg\min_{\boldsymbol{x}_i} \min_{\substack{\mathcal{S} \subset [n] \\ |\mathcal{S}| \geq n - \delta n - 2}} \sum_{j \in \mathcal{S}} \|\boldsymbol{x}_i - \boldsymbol{x}_j\|_2^2.$$

In our experiments, this corresponds to selecting over the $n - \delta n - 2 = 18$ nearest neighbors.

**Centered Clipping (CCLIP).** Centered clipping iteratively averages clipped residuals around a centering point $\boldsymbol{m} \in \mathbb{R}^d$:

$$\mathrm{CCLIP}_{\boldsymbol{m}, L, \tau}(\boldsymbol{x}_1, \ldots, \boldsymbol{x}_n) = \boldsymbol{v}_L,$$

where $\boldsymbol{v}_0 = \boldsymbol{m}$ and

$$\boldsymbol{v}_{\ell+1} = \boldsymbol{v}_\ell + \frac{1}{n} \sum_{i=1}^n (\boldsymbol{x}_i - \boldsymbol{v}_\ell) \min\left(1, \frac{\tau}{\|\boldsymbol{x}_i - \boldsymbol{v}_\ell\|_2}\right), \qquad \ell = 0, \ldots, L - 1.$$

In our experiments, we set $L = 3$ and $\tau = 10.0$, and take $\boldsymbol{m}$ to be the previous round's aggregate.

**Robust Federated Averaging (RFA).** RFA returns the geometric median of the inputs:

$$\mathrm{RFA}(\boldsymbol{x}_1, \ldots, \boldsymbol{x}_n) = \arg\min_{\boldsymbol{v} \in \mathbb{R}^d} \sum_{i=1}^n \|\boldsymbol{v} - \boldsymbol{x}_i\|_2.$$

## B.6. Computing Environment

Experiments ran on NVIDIA A100-SXM4-80GB GPUs (CUDA 12.2) and AMD EPYC 7763 CPUs. Table 4 provides detailed hardware and software specifications.

*Table 4.* Runtime hardware and software.

| | |
|---|---|
| **CPU** | |
| Model name | AMD EPYC 7763 64-Core Processor |
| # CPU(s) | 128 |
| **GPU** | |
| Product Name | NVIDIA A100-SXM4-80GB |
| CUDA Version | 12.2 |
| **PyTorch** | |
| Version | 2.7.1 |

# C. Extended Results

**Per-aggregator curves with Byzantine clients.** This section complemented Figs. 3, 4, 5, 6 by showing training dynamics for the other robust aggregators across the same attacks, data partitions, and optimizers on MNIST (ConvNet) and CIFAR-10 (ResNet-18).

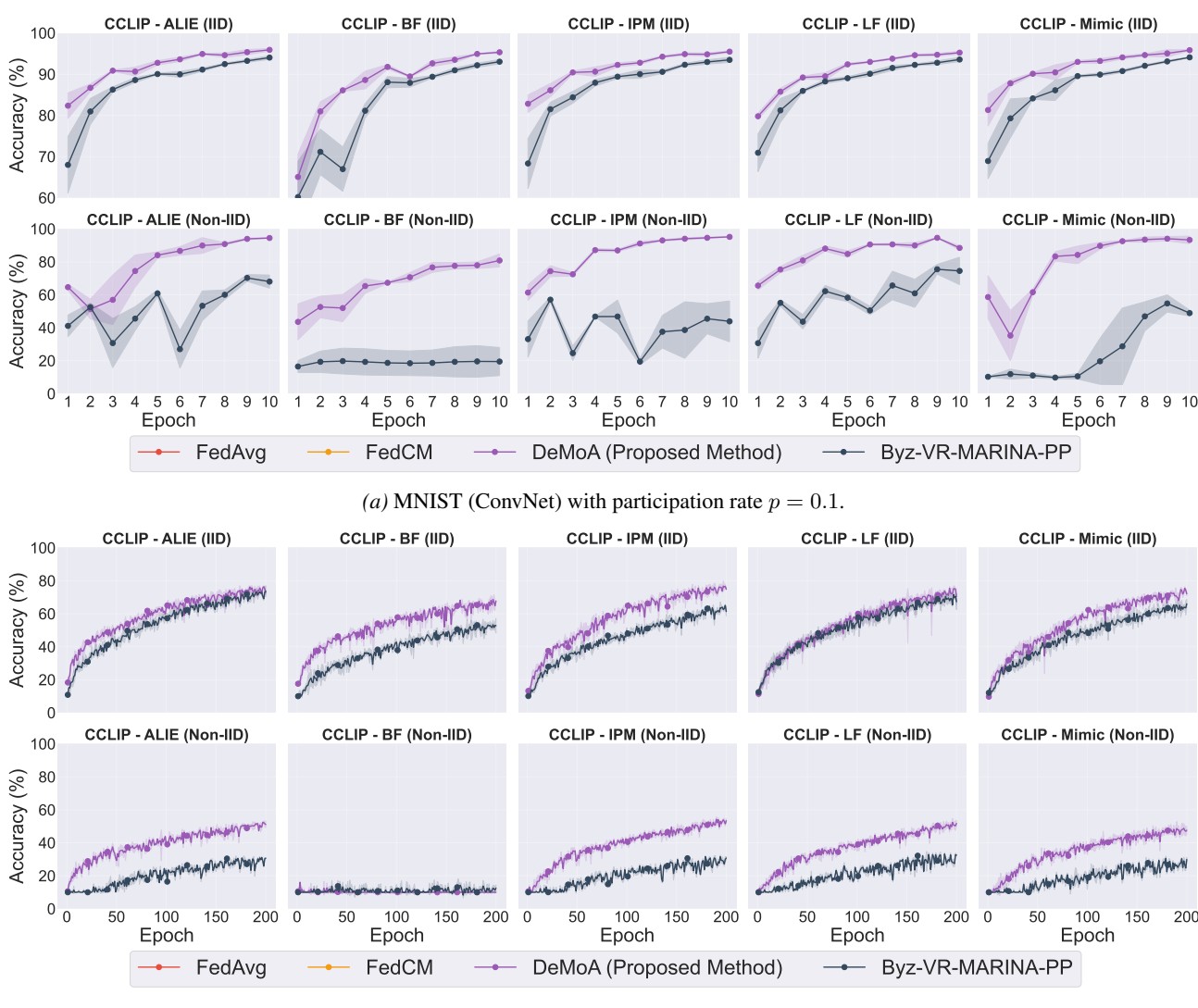

*(a)* MNIST (ConvNet) with participation rate $p = 0.1$.

*(b)* MNIST (ConvNet) with participation rate $p = 0.1$.

*Figure 3.* Training with Byzantine ratio $\delta = 0.2$ using centered clipping (cp) aggregator. The top row shows IID splits and the bottom row shows non-IID splits with bucketing size $s = 2$; from left to right, the columns correspond to ALIE, Bit-Flipping (BF), IPM, Label-Flipping (LF), and Mimic attacks.

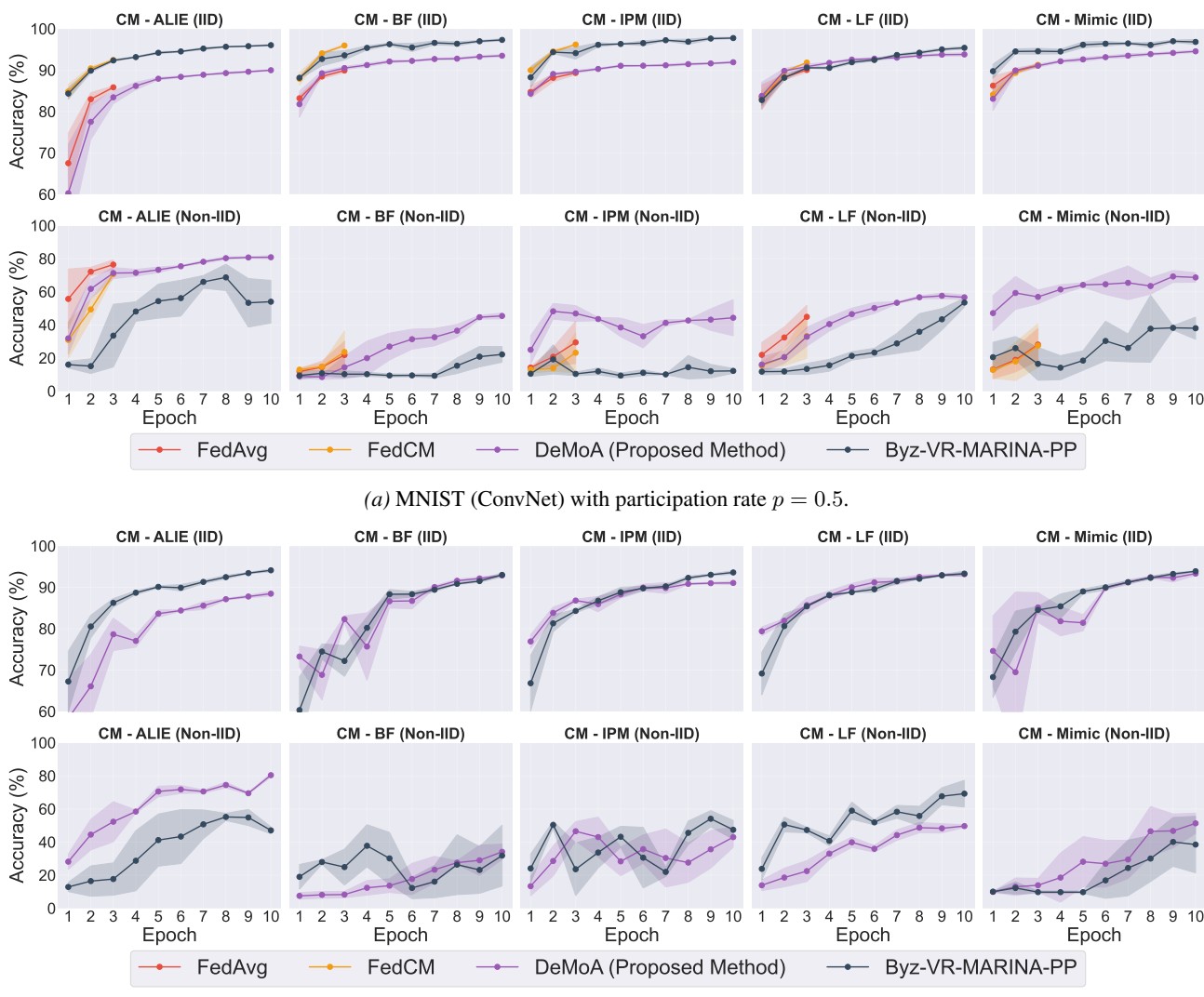

*(a)* MNIST (ConvNet) with participation rate $p = 0.5$.

*(b)* MNIST (ConvNet) with participation rate $p = 0.1$.

*Figure 4.* Training with Byzantine ratio $\delta = 0.2$ using coordinate-wise median (cm) aggregator. The top row shows IID splits and the bottom row shows non-IID splits with bucketing size $s = 2$; from left to right, the columns correspond to ALIE, Bit-Flipping (BF), IPM, Label-Flipping (LF), and Mimic attacks.

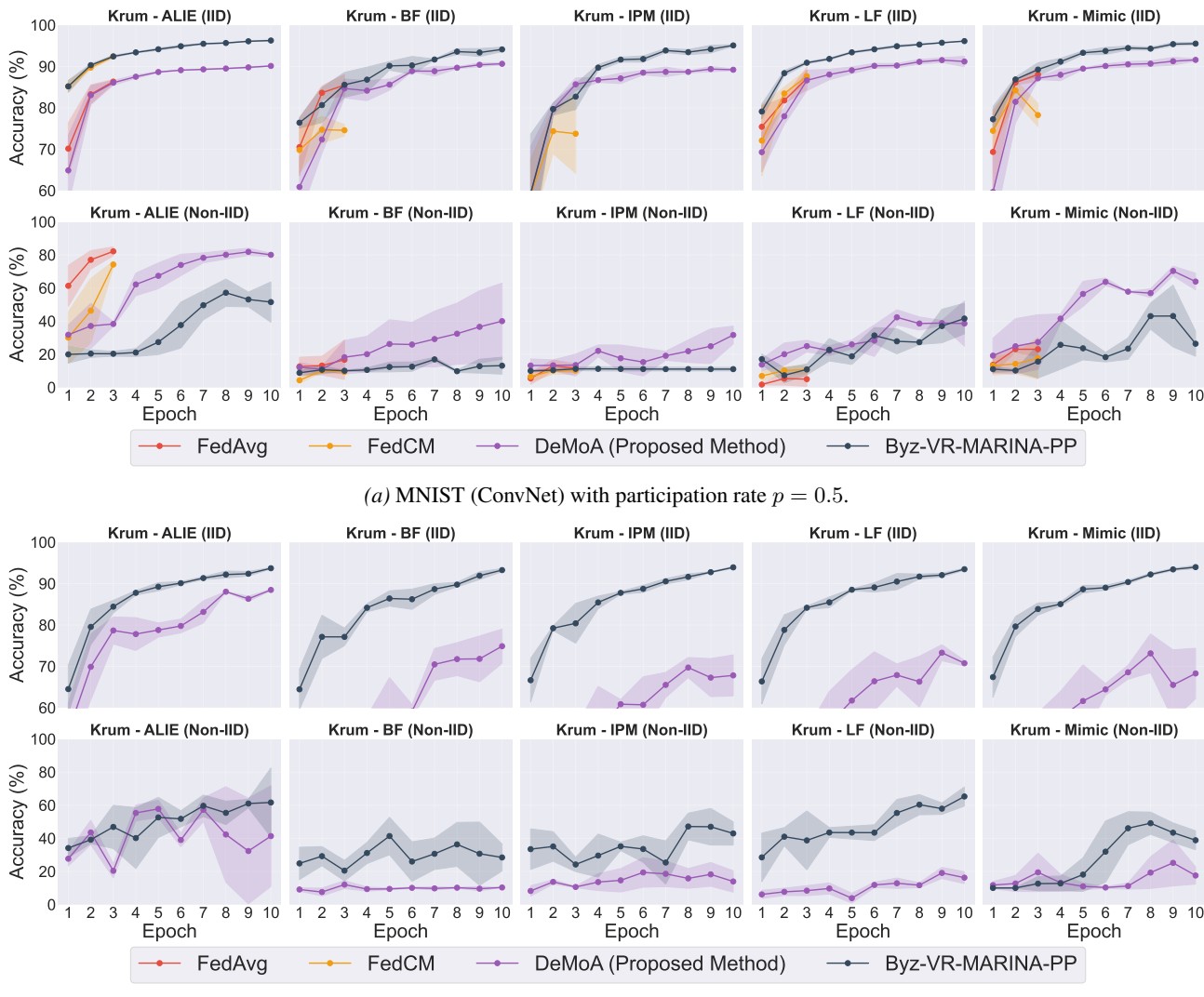

*(a)* MNIST (ConvNet) with participation rate $p = 0.5$.

*(b)* MNIST (ConvNet) with participation rate $p = 0.1$.

*Figure 5.* Training with Byzantine ratio $\delta = 0.2$ using Multi-Krum (krum) aggregator. The top row shows IID splits and the bottom row shows non-IID splits with bucketing size $s = 2$; from left to right, the columns correspond to ALIE, Bit-Flipping (BF), IPM, Label-Flipping (LF), and Mimic attacks.

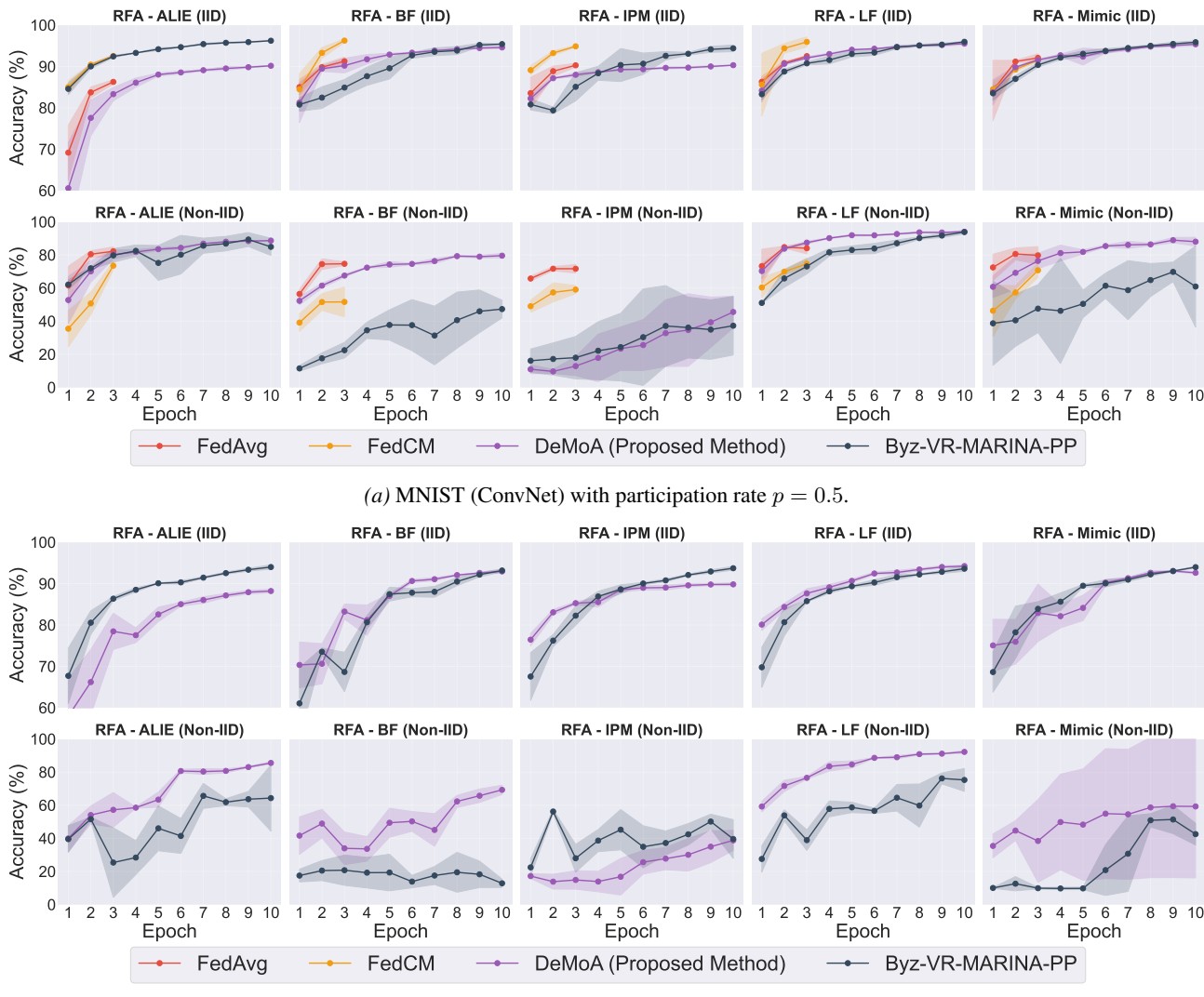

*(a)* MNIST (ConvNet) with participation rate $p = 0.5$.

*(b)* MNIST (ConvNet) with participation rate $p = 0.1$.

*Figure 6.* Training with Byzantine ratio $\delta = 0.2$ using geometric median (`rfa`) aggregator. The top row shows IID splits and the bottom row shows non-IID splits with bucketing size $s = 2$; from left to right, the columns correspond to ALIE, Bit-Flipping (BF), IPM, Label-Flipping (LF), and Mimic attacks.

## D. Useful Lemmas

**Lemma D.1.** *For arbitrary set of $n$ vectors $\left\{ \boldsymbol{a}_i \mid \boldsymbol{a}_i \in \mathbb{R}^d \right\}_{i=1}^n$, it holds that*

$$\left\| \sum_{i=1}^n \mathbf{a}_i \right\|^2 \leq n \sum_{i=1}^n \|\mathbf{a}_i\|^2 .$$

**Lemma D.2.** *For given two vectors $\mathbf{a}, \mathbf{b} \in \mathbb{R}^d$, it holds that*

$$2\langle \mathbf{a}, \mathbf{b} \rangle \leq \gamma \|\mathbf{a}\|^2 + \gamma^{-1} \|\mathbf{b}\|^2, \quad \forall \gamma > 0$$

**Lemma D.3** (Young's inequality)**.** *For given two vectors $\mathbf{a}, \mathbf{b} \in \mathbb{R}^d$, it holds that*

$$\|\mathbf{a} + \mathbf{b}\|^2 \leq (1 + \alpha)\|\mathbf{a}\|^2 + \left(1 + \alpha^{-1}\right) \|\mathbf{b}\|^2, \quad \forall \alpha > 0.$$

## E. Analysis DeMoA under Heterogenous Data

The delayed momentum aggregation principle for the Bernoulli sampling is mathematically equivalent to the following update rule. For each client $i$ at iteration $t$,

$$\boldsymbol{m}_i^t = \begin{cases} (1 - \alpha_t p_t) \, \boldsymbol{m}_i^{t-1} + \alpha_t \nabla f_i(\boldsymbol{x}^{t-1}; \xi_i^t), & \text{if } r_i^t = 1, \\ (1 - \alpha_t p_t) \, \boldsymbol{m}_i^{t-1}, & \text{otherwise,} \end{cases}$$
$$= (1 - \alpha_t p_t)\boldsymbol{m}_i^{t-1} + \alpha_t r_i^t \nabla f_i(\boldsymbol{x}^{t-1}; \xi_i^t).$$

where $r_i^t \sim \mathrm{Ber}(p_t)$ and $\xi_i^t \sim \mathcal{D}_i$. For the following analysis, we set $\alpha_t = \alpha$, $p_t = p$ with $\alpha, p \in [0, 1]$ for $t \geq 2$ and $\alpha_1 = 1$, $p_1 = 1$ for initialization. Define

$$\boldsymbol{m}^t := \mathrm{Agg}(\boldsymbol{m}_1^t, \boldsymbol{m}_2^t, \ldots, \boldsymbol{m}_n^t), \qquad \bar{\boldsymbol{m}}^t := \frac{1}{G} \sum_{i \in \mathcal{G}} \boldsymbol{m}_i^t.$$

The global model is then updated via gradient descent as

$$\boldsymbol{x}^t = \boldsymbol{x}^{t-1} - \eta \boldsymbol{m}^t.$$

Instead of considering $\boldsymbol{m}^t$ directly, we decompose errors into two parts: the aggregation error part $\mathbb{E}\|\boldsymbol{m}^t - \bar{\boldsymbol{m}}^t\|^2$ and the true momentum deviation part $\mathbb{E}\|\bar{\boldsymbol{m}}^t - \nabla f(\boldsymbol{x}^{t-1})\|^2$.

**Lemma E.1** (Descent Lemma)**.** *Let $\alpha, p \in [0, 1]$, $t \geq 2$, and $\eta \leq 1/L$. Suppose that $f$ satisfies Assumption 2.1 (i.e., $f$ is $L$-smooth). Then, for any $t \geq 1$ and given $\boldsymbol{x}^{t-1}$, it holds that*

$$\mathbb{E}\big[f(\boldsymbol{x}^t) \mid \mathcal{F}_{t-1}\big] \leq f(\boldsymbol{x}^{t-1}) - \frac{\eta}{2} \|\nabla f(\boldsymbol{x}^{t-1})\|^2 + \eta \, \mathbb{E}\|\bar{\boldsymbol{m}}^t - \boldsymbol{m}^t\|^2 + \eta \, \mathbb{E}\|\bar{\boldsymbol{m}}^t - \nabla f(\boldsymbol{x}^{t-1})\|^2,$$

*where $\mathcal{F}_{t-1}$ denotes the $\sigma$-algebra generated by all the randomness up to iteration $t - 1$.*

*Proof.* Letting $\eta \leq \frac{1}{L}$, we obtain

$$f(\boldsymbol{x}^t) = f(\boldsymbol{x}^{t-1} - \eta \boldsymbol{m}^t)$$
$$\leq f(\boldsymbol{x}^{t-1}) - \eta \langle \boldsymbol{m}^t, \nabla f(\boldsymbol{x}^{t-1}) \rangle + \frac{L\eta^2}{2} \|\boldsymbol{m}^t\|^2$$
$$\leq f(\boldsymbol{x}^{t-1}) - \eta \langle \boldsymbol{m}^t, \nabla f(\boldsymbol{x}^{t-1}) \rangle + \frac{\eta}{2} \|\boldsymbol{m}^t\|^2$$
$$= f(\boldsymbol{x}^{t-1}) - \frac{\eta}{2} \|\nabla f(\boldsymbol{x}^{t-1})\|^2 + \frac{\eta}{2} \|\nabla f(\boldsymbol{x}^{t-1}) - \boldsymbol{m}^t\|^2$$
$$\leq f(\boldsymbol{x}^{t-1}) - \frac{\eta}{2} \|\nabla f(\boldsymbol{x}^{t-1})\|^2 + \eta \|\nabla f(\boldsymbol{x}^{t-1}) - \bar{\boldsymbol{m}}^t\|^2 + \eta \|\bar{\boldsymbol{m}}^t - \boldsymbol{m}^t\|^2,$$

where the first inequality follows from Assumption 2.1 and Taylor expansion. Taking the expectation on both sides yields the desired result. $\square$

### E.1. Aggregation Error

**Lemma E.2** (Aggregation Error). *Let $\{m_i^t\}_{i\in\mathcal{G}}$ be the set of local momentum and $\bar{m}^t := \frac{1}{G}\sum_{i\in\mathcal{G}} m_i^t$. Define*

$$\rho_t := \frac{1}{G(G-1)}\sum_{i,j\in\mathcal{G}} \mathbb{E}\|m_i^t - m_j^t\|^2.$$

*Then, under the robust aggregation rule 2.4 and Assumptions 2.2 and 2.3, for $t \geq 2$, it holds that*

$$\mathbb{E}\|m^t - \bar{m}^t\|^2 \leq c\delta\,\rho_t \leq 6c\delta\varrho_t,$$

*where $\varrho_t := \frac{1}{G}\sum_{i\in\mathcal{G}}\mathbb{E}\|m_i^t - \mathbb{E}_t[m_i^t]\|^2 + \mathbb{E}\|\mathbb{E}_t[\bar{m}^t] - \bar{m}^t\|^2 + \mathbb{E}\|\mathbb{E}_t[m_i^t] - \mathbb{E}_t[\bar{m}^t]\|^2$, which has a contraction*

$$\varrho_t \leq (1-\alpha p)\varrho_{t-1} + 2\alpha p(\zeta^2 + B^2\mathbb{E}\|\nabla f(x^{t-1})\|^2) + 2\alpha^2 p(1-p)\mathbb{E}\|\nabla f(x^{t-1})\|^2 + 2p\alpha^2\sigma^2.$$

*Moreover, for $t = 1$,*

$$\mathbb{E}\|m^1 - \bar{m}^1\|^2 \leq 6\sigma^2 + 3\zeta^2 + 3B^2\|\nabla f(x^0)\|^2$$

Throughout this subsection, $\mathbb{E}_t[\cdot]$ denotes the expectation with respect to all randomness up to time $t$, i.e., $\mathbb{E}_t[\cdot] = \mathbb{E}_{\xi_1^t,\ldots,\xi_n^t,\xi_1^{t-1},\ldots,\xi_n^{t-1},\ldots,\xi_1^0,\ldots,\xi_n^0,r_1^t,\ldots,r_n^t,r_1^{t-1},\ldots,r_n^{t-1},\ldots,r_1^0,\ldots,r_n^0}[\cdot]$.

**Proposition E.3** (Local variance term). *For any $i \in \mathcal{G}$ and $t \geq 2$,*

$$\mathbb{E}\|m_i^t - \mathbb{E}_t[m_i^t]\|^2 \leq (1-\alpha p)^2\mathbb{E}\|m_i^{t-1} - \mathbb{E}_t[m_i^{t-1}]\|^2 + p\alpha^2\sigma^2 + \alpha^2 p(1-p)\mathbb{E}\|\nabla f_i(x^{t-1})\|^2.$$

*Proof.* Due to the unbiasedness of $r_i^t\nabla f_i(x^{t-1};\xi_i^t)$, i.e., $\mathbb{E}_t[r_i^t\nabla f_i(x^{t-1};\xi_i^t)] = p\nabla f_i(x^{t-1})$,

$$\begin{aligned}
\mathbb{E}\|m_i^t - \mathbb{E}_t[m_i^t]\|^2 &= \mathbb{E}\left\|(1-\alpha p)(m_i^{t-1} - \mathbb{E}_t[m_i^{t-1}]) + \alpha(r_i^t\nabla f_i(x^{t-1};\xi_i^t) - p\nabla f_i(x^{t-1}))\right\|^2 \\
&= (1-\alpha p)^2\mathbb{E}\|m_i^{t-1} - \mathbb{E}_t[m_i^{t-1}]\|^2 + \alpha^2\mathbb{E}\|r_i^t\nabla f_i(x^{t-1};\xi_i^t) - p\nabla f_i(x^{t-1})\|^2.
\end{aligned}$$

Using Assumption 2.2 on the last term yields

$$\begin{aligned}
&\mathbb{E}\|r_i^t\nabla f_i(x^{t-1};\xi_i^t) - p\nabla f_i(x^{t-1})\|^2 \\
&= p\mathbb{E}\|\nabla f_i(x^{t-1};\xi_i^t) - p\nabla f_i(x^{t-1})\|^2 + (1-p)\mathbb{E}\|p\nabla f_i(x^{t-1})\|^2 \\
&= p\mathbb{E}\|\nabla f_i(x^{t-1};\xi_i^t) - \nabla f_i(x^{t-1}) + (1-p)\nabla f_i(x^{t-1})\|^2 + (1-p)\mathbb{E}\|p\nabla f_i(x^{t-1})\|^2 \\
&\leq p\sigma^2 + p(1-p)\mathbb{E}\|\nabla f_i(x^{t-1})\|^2,
\end{aligned}$$

which gives the claim. $\qquad\square$

**Proposition E.4** (Averaged variance term). *For $t \geq 2$,*

$$\mathbb{E}\|\mathbb{E}_t[\bar{m}^t] - \bar{m}^t\|^2 \leq (1-\alpha p)^2\mathbb{E}\|\bar{m}^{t-1} - \mathbb{E}_{t-1}[\bar{m}^{t-1}]\|^2 + \frac{\alpha^2 p(1-p)}{G}\cdot\frac{1}{G}\sum_{i\in\mathcal{G}}\mathbb{E}\|\nabla f_i(x^{t-1})\|^2 + \frac{p\alpha^2\sigma^2}{G}.$$

*Proof.* Independence implies $\mathbb{E}\left[\frac{1}{G}\sum_{i\in\mathcal{G}} r_i^t\nabla f_i(x^{t-1};\xi_i^t)\right] = p\nabla f(x^{t-1})$, so

$$\begin{aligned}
&\mathbb{E}\|\mathbb{E}_t[\bar{m}^t] - \bar{m}^t\|^2 \\
&= \mathbb{E}\left\|(1-\alpha p)(\bar{m}^{t-1} - \mathbb{E}_{t-1}[\bar{m}^{t-1}]) + \alpha\left(\left(\frac{1}{G}\sum_{i\in\mathcal{G}} r_i^t\nabla f_i(x^{t-1};\xi_i^t)\right) - p\nabla f(x^{t-1})\right)\right\|^2 \\
&= (1-\alpha p)^2\mathbb{E}\|\bar{m}^{t-1} - \mathbb{E}_{t-1}[\bar{m}^{t-1}]\|^2 + \alpha^2\mathbb{E}\left\|\left(\frac{1}{G}\sum_{i\in\mathcal{G}} r_i^t\nabla f_i(x^{t-1};\xi_i^t)\right) - p\nabla f(x^{t-1})\right\|^2.
\end{aligned}$$

Further, using independence, the second term in the last equation can be decomposed as

$$\mathbb{E}\left\|\left(\frac{1}{G}\sum_{i\in\mathcal{G}}r_i^t\nabla f_i(\boldsymbol{x}^{t-1};\xi_i^t)\right)\pm\left(\frac{1}{G}\sum_{i\in\mathcal{G}}r_i^t\nabla f_i(\boldsymbol{x}^{t-1})\right)-p\nabla f(\boldsymbol{x}^{t-1})\right\|^2$$

$$=\mathbb{E}\left\|\frac{1}{G}\sum_{i\in\mathcal{G}}r_i^t(\nabla f_i(\boldsymbol{x}^{t-1};\xi_i^t)-\nabla f_i(\boldsymbol{x}^{t-1}))\right\|^2+\mathbb{E}\left\|\frac{1}{G}\sum_{i\in\mathcal{G}}(r_i^t-p)\nabla f_i(\boldsymbol{x}^{t-1})\right\|^2.$$

By Assumption 2.2, the first term is at most $p\sigma^2/G$. For the second,

$$\mathbb{E}\left\|\frac{1}{G}\sum_{i\in\mathcal{G}}(r_i^t-p)\nabla f_i(\boldsymbol{x}^{t-1})\right\|^2=\frac{1}{G^2}\sum_{i\in\mathcal{G}}\mathbb{E}(r_i^t-p)^2\|\nabla f_i(\boldsymbol{x}^{t-1})\|^2$$

$$\leq\frac{p(1-p)}{G}\cdot\frac{1}{G}\sum_{i\in\mathcal{G}}\mathbb{E}\|\nabla f_i(\boldsymbol{x}^{t-1})\|^2,$$

because the cross terms vanish. Combining the pieces proves the claim. $\qquad\square$

**Proposition E.5** (Client drift term). *For any $i\in\mathcal{G}$ and $t\geq 2$,*

$$\mathbb{E}\|\mathbb{E}_t[\boldsymbol{m}_i^t]-\mathbb{E}_t[\bar{\boldsymbol{m}}^t]\|^2\leq(1-\alpha p)\mathbb{E}\|\mathbb{E}_{t-1}[\boldsymbol{m}_i^{t-1}]-\mathbb{E}_{t-1}[\bar{\boldsymbol{m}}^{t-1}]\|^2+\alpha p\mathbb{E}\|\nabla f_i(\boldsymbol{x}^{t-1})-\nabla f(\boldsymbol{x}^{t-1})\|^2.$$

*Proof.* By Jensen's inequality,

$$\mathbb{E}\|\mathbb{E}_t[\boldsymbol{m}_i^t]-\mathbb{E}_t[\bar{\boldsymbol{m}}^t]\|^2=\mathbb{E}\|(1-\alpha p)(\mathbb{E}_{t-1}[\boldsymbol{m}_i^{t-1}]-\mathbb{E}_{t-1}[\bar{\boldsymbol{m}}^{t-1}])+\alpha p(\nabla f_i(\boldsymbol{x}^{t-1})-\nabla f(\boldsymbol{x}^{t-1}))\|^2$$

$$\leq(1-\alpha p)\mathbb{E}\|\mathbb{E}_{t-1}[\boldsymbol{m}_i^{t-1}]-\mathbb{E}_{t-1}[\bar{\boldsymbol{m}}^{t-1}]\|^2+\alpha p\mathbb{E}\|\nabla f_i(\boldsymbol{x}^{t-1})-\nabla f(\boldsymbol{x}^{t-1})\|^2.$$

$\qquad\square$

**Proposition E.6** (Contraction of $\varrho_t$). *Under Assumptions 2.2 and 2.3, for $t\geq 2$,*

$$\varrho_t\leq(1-\alpha p)\varrho_{t-1}+3\alpha p(\zeta^2+B^2\mathbb{E}\|\nabla f(\boldsymbol{x}^{t-1})\|^2)+2\alpha^2 p(1-p)\mathbb{E}\|\nabla f(\boldsymbol{x}^{t-1})\|^2+2p\alpha^2\sigma^2.$$

*Proof.* Applying Propositions E.3, E.4, and E.5 to the definition of $\varrho_t$ yields, for $t\geq 2$,

$$\varrho_t\leq(1-\alpha p)\varrho_{t-1}+\frac{2\alpha^2 p(1-p)}{G}\sum_{i\in\mathcal{G}}\mathbb{E}\|\nabla f_i(\boldsymbol{x}^{t-1})\|^2+\frac{\alpha p}{G}\sum_{i\in\mathcal{G}}\mathbb{E}\|\nabla f_i(\boldsymbol{x}^{t-1})-\nabla f(\boldsymbol{x}^{t-1})\|^2$$

$$+p\alpha^2\sigma^2+\frac{p\alpha^2\sigma^2}{G}.$$

Since $\frac{1}{G}\sum_{i\in\mathcal{G}}\|\nabla f_i(\boldsymbol{x}^{t-1})\|^2=\frac{1}{G}\sum_{i\in\mathcal{G}}\|\nabla f_i(\boldsymbol{x}^{t-1})-\nabla f(\boldsymbol{x}^{t-1})\|^2+\|\nabla f(\boldsymbol{x}^{t-1})\|^2$,

$$\varrho_t\leq(1-\alpha p)\varrho_{t-1}+\left(\frac{2\alpha^2 p(1-p)}{G}+\frac{\alpha p}{G}\right)\sum_{i\in\mathcal{G}}\mathbb{E}\|\nabla f_i(\boldsymbol{x}^{t-1})-\nabla f(\boldsymbol{x}^{t-1})\|^2$$

$$+2\alpha^2 p(1-p)\mathbb{E}\|\nabla f(\boldsymbol{x}^{t-1})\|^2+p\alpha^2\sigma^2+\frac{p\alpha^2\sigma^2}{G}$$

$$\leq(1-\alpha p)\varrho_{t-1}+\frac{3\alpha p}{G}\sum_{i\in\mathcal{G}}\mathbb{E}\|\nabla f_i(\boldsymbol{x}^{t-1})-\nabla f(\boldsymbol{x}^{t-1})\|^2$$

$$+2\alpha^2 p(1-p)\mathbb{E}\|\nabla f(\boldsymbol{x}^{t-1})\|^2+2p\alpha^2\sigma^2$$

$$\leq(1-\alpha p)\varrho_{t-1}+3\alpha p(\zeta^2+B^2\mathbb{E}\|\nabla f(\boldsymbol{x}^{t-1})\|^2)$$

$$+2\alpha^2 p(1-p)\mathbb{E}\|\nabla f(\boldsymbol{x}^{t-1})\|^2+2p\alpha^2\sigma^2.$$

$\qquad\square$

**Proposition E.7** (Initialization). *If $p_t = \alpha_t = 1$, then*

$$\mathbb{E}\|\boldsymbol{m}^1 - \bar{\boldsymbol{m}}^1\|^2 \le 6\sigma^2 + 3\zeta^2 + 3B^2\|\nabla f(\boldsymbol{x}^0)\|^2.$$

*Proof.* Since $\boldsymbol{m}_i^1 = \nabla f(\boldsymbol{x}^0; \xi_i^1)$,

$$\frac{1}{G}\sum_{i\in\mathcal{G}}\mathbb{E}\|\boldsymbol{m}_i^1 - \bar{\boldsymbol{m}}^1\|^2 \le \frac{3}{G}\sum_{i\in\mathcal{G}}\mathbb{E}\|\boldsymbol{m}_i^1 - \nabla f_i(\boldsymbol{x}^0)\|^2 + 3\mathbb{E}\|\nabla f(\boldsymbol{x}^0) - \bar{\boldsymbol{m}}^1\|^2 + \frac{3}{G}\sum_{i\in\mathcal{G}}\mathbb{E}\|\nabla f_i(\boldsymbol{x}^0) - \nabla f(\boldsymbol{x}^0)\|^2$$

$$\le 3\sigma^2 + \frac{3\sigma^2}{n} + 3(\zeta^2 + B^2\|\nabla f(\boldsymbol{x}^0)\|^2)$$

$$\le 6\sigma^2 + 3\zeta^2 + 3B^2\|\nabla f(\boldsymbol{x}^0)\|^2.$$

$\square$

*Proof of Lemma E.2.* By definition of the $(\delta, c)$-robust aggregator (Assumption 2.4), $\mathbb{E}\|\boldsymbol{m}^t - \bar{\boldsymbol{m}}^t\|^2 \le c\delta\,\rho_t$ where $\rho_t$ is defined above. Lemma D.1 implies $\rho_t \le 6\varrho_t$. The recursion for $\varrho_t$ for $t \ge 2$ is Proposition E.6, and Proposition E.7 gives the base case $t = 1$. $\square$

## E.2. Error Analysis

**Lemma E.8** (Error bound). *Given* Agg *satisfies the $(\delta, c)$-robust aggregator 2.4, and under assumption 2.3, for $t \ge 2$, we have:*

$$\mathbb{E}\|\bar{\boldsymbol{m}}^t - \nabla f(\boldsymbol{x}^{t-1})\|^2 \le \left(1 - \frac{2\alpha p}{5}\right)\mathbb{E}\|\bar{\boldsymbol{m}}^{t-1} - \nabla f(\boldsymbol{x}^{t-2})\|^2$$

$$+ \frac{9L^2\eta^2}{\alpha p}\mathbb{E}\|\boldsymbol{m}^{t-1} - \bar{\boldsymbol{m}}^{t-1}\|^2 + \frac{9L^2\eta^2}{\alpha p}\mathbb{E}\|\nabla f(\boldsymbol{x}^{t-2})\|^2$$

$$+ \alpha^2\left(\frac{p(1-p)}{G}(1+B^2)\mathbb{E}\|\nabla f(\boldsymbol{x}^{t-1})\|^2 + \frac{p\sigma^2}{G} + \frac{p(1-p)}{G}\zeta^2\right)$$

*Moreover, for $t = 1$,*

$$\mathbb{E}\|\bar{m}^1 - \nabla f(\boldsymbol{x}^0)\|^2 = \mathbb{E}\left\|\frac{1}{G}\sum_{i\in\mathcal{G}}\nabla f(\boldsymbol{x}^0; \xi_i^1) - \nabla f(\boldsymbol{x}^0)\right\|^2$$

$$\le \frac{\sigma^2}{G}.$$

*Proof.* To upper bound the error term $\mathbb{E}\|\bar{\boldsymbol{m}}^t - \nabla f(\boldsymbol{x}^{t-1})\|^2$, for $t \ge 2$,

$$\mathbb{E}\|\bar{\boldsymbol{m}}^t - \nabla f(\boldsymbol{x}^{t-1})\|^2 = \mathbb{E}\|\frac{1}{G}\sum_{i\in\mathcal{G}}(1-\alpha p)\boldsymbol{m}_i^{t-1} + \alpha r_i^t\nabla f_i(\boldsymbol{x}^{t-1}; \xi_i^t) - \nabla f(\boldsymbol{x}^{t-1})\|^2$$

$$= \mathbb{E}\|(1-\alpha p)(\bar{\boldsymbol{m}}^{t-1} - \nabla f(\boldsymbol{x}^{t-1})) + \frac{\alpha}{G}\sum_{i\in\mathcal{G}}(r_i^t\nabla f_i(\boldsymbol{x}^{t-1}; \xi_i^t) - p\nabla f(\boldsymbol{x}^{t-1}))\|^2$$

$$= \mathbb{E}\|(1-\alpha p)(\bar{\boldsymbol{m}}^{t-1} - \nabla f(\boldsymbol{x}^{t-1})\|^2 + \mathbb{E}\|\frac{\alpha}{G}\sum_{i\in\mathcal{G}}(r_i^t\nabla f_i(\boldsymbol{x}^{t-1}; \xi_i^t) - p\nabla f(\boldsymbol{x}^{t-1}))\|^2$$

The second and third equalities follow from the unbiasedness and independence of $r_i^t$ and stochastic gradients. For the

second term, since for any $i \in \mathcal{G}$,

$$\mathbb{E}\|\frac{1}{G}\sum_{i\in\mathcal{G}}r_i^t\nabla f_i(\boldsymbol{x}^{t-1};\xi_i^t) - p\nabla f(\boldsymbol{x}^{t-1})\|^2 = \mathbb{E}\|\frac{1}{G}\sum_{i\in\mathcal{G}}r_i^t(\nabla f_i(\boldsymbol{x}^{t-1};\xi_i^t) - \nabla f(\boldsymbol{x}^{t-1})) + (r_i^t - p)\nabla f(\boldsymbol{x}^{t-1})\|^2$$

$$= \mathbb{E}\|\frac{1}{G}\sum_{i\in\mathcal{G}}r_i^t(\nabla f_i(\boldsymbol{x}^{t-1};\xi_i^t) - \nabla f(\boldsymbol{x}^{t-1}))\|^2 + \mathbb{E}\|\frac{1}{G}\sum_{i\in\mathcal{G}}(r_i^t - p)\nabla f(\boldsymbol{x}^{t-1})\|^2$$

$$= \mathbb{E}\|\frac{1}{G}\sum_{i\in\mathcal{G}}r_i^t(\nabla f_i(\boldsymbol{x}^{t-1};\xi_i^t) - \nabla f(\boldsymbol{x}^{t-1}))\|^2 + \mathbb{E}|\frac{1}{G}\sum_{i\in\mathcal{G}}(r_i^t - p)|^2\mathbb{E}\|\nabla f(\boldsymbol{x}^{t-1})\|^2$$

$$\leq \mathbb{E}\|\frac{1}{G}\sum_{i\in\mathcal{G}}r_i^t(\nabla f_i(\boldsymbol{x}^{t-1};\xi_i^t) - \nabla f(\boldsymbol{x}^{t-1}))\|^2 + \frac{p(1-p)}{G}\mathbb{E}\|\nabla f(\boldsymbol{x}^{t-1})\|^2$$

The first term can further be upper bounded as:

$$\mathbb{E}\|\frac{1}{G}\sum_{i\in\mathcal{G}}r_i^t(\nabla f_i(\boldsymbol{x}^{t-1};\xi_i^t) - \nabla f_i(\boldsymbol{x}^{t-1})) + r_i^t(\nabla f_i(\boldsymbol{x}^{t-1}) - \nabla f(\boldsymbol{x}^{t-1}))\|^2$$

$$= \mathbb{E}\|\frac{1}{G}\sum_{i\in\mathcal{G}}r_i^t(\nabla f_i(\boldsymbol{x}^{t-1};\xi_i^t) - \nabla f_i(\boldsymbol{x}^{t-1}))\|^2 + \mathbb{E}\|\frac{1}{G}\sum_{i\in\mathcal{G}}r_i^t(\nabla f_i(\boldsymbol{x}^{t-1}) - \nabla f(\boldsymbol{x}^{t-1}))\|^2$$

$$\leq \frac{p\sigma^2}{G} + \underbrace{\|\mathbb{E}[\frac{1}{G}\sum_{i\in\mathcal{G}}r_i^t(\nabla f_i(\boldsymbol{x}^{t-1}) - \nabla f(\boldsymbol{x}^{t-1}))]\|^2}_{0} + \mathrm{Var}[\frac{1}{G}\sum_{i\in\mathcal{G}}r_i^t(\nabla f_i(\boldsymbol{x}^{t-1}) - \nabla f(\boldsymbol{x}^{t-1}))]$$

$$= \frac{p\sigma^2}{G} + \frac{1}{G^2}\sum_{i\in\mathcal{G}}\mathrm{Var}[r_i^t(\nabla f_i(\boldsymbol{x}^{t-1}) - \nabla f(\boldsymbol{x}^{t-1}))]$$

$$\leq \frac{p\sigma^2}{G} + \frac{1}{G^2}\sum_{i\in\mathcal{G}}\mathbb{E}\|r_i^t(\nabla f_i(\boldsymbol{x}^{t-1}) - \nabla f(\boldsymbol{x}^{t-1}))\|^2 - \|p(\nabla f_i(\boldsymbol{x}^{t-1}) - \nabla f(\boldsymbol{x}^{t-1}))\|^2$$

$$\leq \frac{p\sigma^2}{G} + \frac{p(1-p)}{G}\cdot\frac{1}{G}\sum_{i\in\mathcal{G}}\mathbb{E}\|\nabla f_i(\boldsymbol{x}^{t-1}) - \nabla f(\boldsymbol{x}^{t-1})\|^2$$

$$\leq \frac{p\sigma^2}{G} + \frac{p(1-p)}{G}(\zeta^2 + B^2\mathbb{E}\|\nabla f(\boldsymbol{x}^{t-1})\|^2)$$

For the error term, now we have:

$$\mathbb{E}\|\bar{m}^t - \nabla f(\boldsymbol{x}^{t-1})\|^2 \leq (1-\alpha p)^2\mathbb{E}\|\bar{\boldsymbol{m}}^{t-1} - \nabla f(\boldsymbol{x}^{t-1})\|^2$$
$$+ \alpha^2\left(\frac{p(1-p)}{G}(1+B^2)\mathbb{E}\|\nabla f(\boldsymbol{x}^{t-1})\|^2 + \frac{p\sigma^2}{G} + \frac{p(1-p)}{G}\zeta^2\right)$$

Since

$$(1-\alpha p)\mathbb{E}\|\bar{\boldsymbol{m}}^{t-1} - \nabla f(\boldsymbol{x}^{t-1})\|^2 \leq (1-\alpha p/2)\mathbb{E}\|\bar{\boldsymbol{m}}^{t-1} - \nabla f(\boldsymbol{x}^{t-2})\|^2 + (1-\alpha p)(1+\frac{2}{\alpha p})\mathbb{E}\|\nabla f(\boldsymbol{x}^{t-2}) - \nabla f(\boldsymbol{x}^{t-1})\|^2$$

$$\leq (1-\alpha p/2)\mathbb{E}\|\bar{\boldsymbol{m}}^{t-1} - \nabla f(\boldsymbol{x}^{t-2})\|^2 + \frac{3L^2\eta^2}{\alpha p}\mathbb{E}\|\boldsymbol{m}^{t-1} \pm \bar{\boldsymbol{m}}^{t-1} \pm \nabla f(\boldsymbol{x}^{t-2})\|^2$$

$$\leq \left(1 - \frac{\alpha p}{2} + \frac{9L^2\eta^2}{\alpha p}\right)\mathbb{E}\|\bar{\boldsymbol{m}}^{t-1} - \nabla f(\boldsymbol{x}^{t-2})\|^2 + \frac{9L^2\eta^2}{\alpha p}\mathbb{E}\|\boldsymbol{m}^{t-1} - \bar{\boldsymbol{m}}^{t-1}\|^2 + \frac{9L^2\eta^2}{\alpha p}\mathbb{E}\|\nabla f(\boldsymbol{x}^{t-2})\|^2.$$

Choosing $\alpha$ so that

$$1 - \frac{\alpha p}{2} + \frac{9L^2\eta^2}{\alpha p} \leq 1 - \frac{2\alpha p}{5} \implies \alpha \geq \sqrt{90}L\eta/p,$$

Finally we have,

$$\mathbb{E}\|\bar{m}^t - \nabla f(\boldsymbol{x}^{t-1})\|^2 \leq \left(1 - \frac{2\alpha p}{5}\right)\mathbb{E}\|\bar{m}^{t-1} - \nabla f(\boldsymbol{x}^{t-2})\|^2$$
$$+ \frac{9L^2\eta^2}{\alpha p}\mathbb{E}\|\boldsymbol{m}^{t-1} - \bar{\boldsymbol{m}}^{t-1}\|^2 + \frac{9L^2\eta^2}{\alpha p}\mathbb{E}\|\nabla f(\boldsymbol{x}^{t-2})\|^2$$
$$+ \alpha^2\left(\frac{p(1-p)}{G}(1+B^2)\mathbb{E}\|\nabla f(\boldsymbol{x}^{t-1})\|^2 + \frac{p\sigma^2}{G} + \frac{p(1-p)}{G}\zeta^2\right)$$

For $t = 1$, since $\alpha_1 = p_1 = 1$,

$$\mathbb{E}\|\bar{\boldsymbol{m}}^1 - \nabla f(\boldsymbol{x}^0)\|^2 = \mathbb{E}\left\|\frac{1}{G}\sum_{i\in\mathcal{G}}\nabla f(\boldsymbol{x}^0;\xi_i^1) - \nabla f(\boldsymbol{x}^0)\right\|^2$$
$$\leq \frac{\sigma^2}{G}.$$

$\square$

### E.3. Final Bound

*Proof of Theorem 3.1.* Define $\bar{e}_t := \mathbb{E}\|\bar{\boldsymbol{m}}^t - \nabla f(\boldsymbol{x}^{t-1})\|^2$. Combining Lemmas E.1, E.2, and E.8, since $\rho_t \leq 6\varrho_t$ for $t \geq 2$, we have

$$\bar{e}_t \leq \left(1 - \frac{2\alpha p}{5}\right)\bar{e}_{t-1} + \frac{54L^2\eta^2}{\alpha p}c\delta\varrho_{t-1} + \frac{9L^2\eta^2}{\alpha p}\mathbb{E}\|\nabla f(\boldsymbol{x}^{t-2})\|^2$$
$$+ \alpha^2\left(\frac{p(1-p)}{G}(1+B^2)\mathbb{E}\|\nabla f(\boldsymbol{x}^{t-1})\|^2 + \frac{p\sigma^2}{G} + \frac{p(1-p)}{G}\zeta^2\right)$$
$$\varrho_t \leq (1-\alpha p)\varrho_{t-1} + 2\alpha p(\zeta^2 + B^2\mathbb{E}\|\nabla f(\boldsymbol{x}^{t-1})\|^2)$$
$$+ 2\alpha^2 p(1-p)\mathbb{E}\|\nabla f(\boldsymbol{x}^{t-1})\|^2 + 2p\alpha^2\sigma^2$$

and for $t = 1$,

$$\bar{e}_1 \leq \frac{\sigma^2}{G}.$$
$$\varrho_1 \leq 2\sigma^2 + \zeta^2 + B^2\|\nabla f(x^0)\|^2$$

Define a Lyapunov function of this form:

$$\Phi_t := \mathbb{E}f(\boldsymbol{x}^t) - f^* + A_\Phi\bar{e}_t + B_\Phi\mathbb{E}\|\nabla f(\boldsymbol{x}^{t-1})\|^2 + 6C_\Phi c\delta\varrho_t, \quad A, B, C > 0.$$

Then, for all $t \geq 2$,

$$
\begin{aligned}
\Phi_t - \Phi_{t-1} &= f(\boldsymbol{x}^t) - f(\boldsymbol{x}^{t-1}) + A_\Phi(\bar{e}^t - \bar{e}^{t-1}) + B_\Phi(\mathbb{E}\|\nabla f(\boldsymbol{x}^{t-1})\|^2 - \mathbb{E}\|\nabla f(\boldsymbol{x}^{t-2})\|^2) + 3C_\Phi c\delta(\varrho_t - \varrho_{t-1}) \\
&\leq (\eta + A_\Phi)\,\bar{e}^t - A_\Phi \bar{e}^{t-1} + (B_\Phi - \eta/2)\,\mathbb{E}\|\nabla f(\boldsymbol{x}^{t-1})\|^2 - B_\Phi \mathbb{E}\|\nabla f(\boldsymbol{x}^{t-2})\|^2 + 3c\delta(\eta + C_\Phi)\varrho_t - 3C_\Phi c\delta \varrho_{t-1} \\
&\leq \left[(\eta + A_\Phi)\left(1 - \frac{2\alpha p}{5}\right) - A_\Phi\right]\bar{e}_{t-1} \\
&\quad + \left[(\eta + A_\Phi)\frac{\alpha^2 p(1-p)}{G}(1 + B^2) + (B_\Phi - \eta/2)\right]\mathbb{E}\|\nabla f(\boldsymbol{x}^{t-1})\|^2 \\
&\quad + \left[(\eta + A_\Phi)\frac{9L^2\eta^2}{\alpha p} - B_\Phi\right]\mathbb{E}\|\nabla f(\boldsymbol{x}^{t-2})\|^2 \\
&\quad + 3c\delta(\eta + C_\Phi)\varrho_t \\
&\quad + \left[(\eta + A_\Phi)\frac{27c\delta L^2\eta^2}{\alpha p} - 3C_\Phi c\delta\right]\varrho_{t-1} \\
&\quad + (\eta + A_\Phi)\left(\frac{p\alpha^2\sigma^2}{G} + \frac{\alpha^2 p(1-p)\zeta^2}{G}\right) \\
&\leq \left[(\eta + A_\Phi)\left(1 - \frac{2\alpha p}{5}\right) - A_\Phi\right]\bar{e}_{t-1} \\
&\quad + \left[(\eta + A_\Phi)\frac{\alpha^2 p(1-p)}{G}(1 + B^2) + (B_\Phi - \eta/2) + 3c\delta(\eta + C_\Phi)(2\alpha p B^2 + 2\alpha^2 p(1-p))\right]\mathbb{E}\|\nabla f(\boldsymbol{x}^{t-1})\|^2 \\
&\quad + \left[(\eta + A_\Phi)\frac{9L^2\eta^2}{\alpha p} - B_\Phi\right]\mathbb{E}\|\nabla f(\boldsymbol{x}^{t-2})\|^2 \\
&\quad + \left[(\eta + A_\Phi)\frac{27c\delta L^2\eta^2}{\alpha p} + 3c\delta(\eta + C_\Phi)(1 - \alpha p) - 3C_\Phi c\delta\right]\varrho_{t-1} \\
&\quad + (\eta + A_\Phi)\left(\frac{p\alpha^2\sigma^2}{G} + \frac{\alpha^2 p(1-p)\zeta^2}{G}\right) + 3c\delta(\eta + C_\Phi)(2\alpha p\zeta^2 + 2p\alpha^2\sigma^2)
\end{aligned}
$$

Making the coefficients of $\bar{e}_{t-1}, \mathbb{E}\|\nabla f(\boldsymbol{x}^{t-2})\|^2$ and $\varrho_{t-1}$ to be zero, we have the following solutions for $A_\Phi, B_\Phi, C_\Phi$:

$$
A_\Phi = \frac{5\eta}{2\alpha p} - \eta
$$

$$
B_\Phi = \frac{45L^2\eta^3}{2\alpha^2 p^2}
$$

$$
C_\Phi = \frac{45L^2\eta^3}{2\alpha^3 p^3} + \frac{\eta}{\alpha p} - \eta
$$

To make gradient term $\mathbb{E}\|\nabla f(\boldsymbol{x}^{t-1})\|^2$ to be negative, it suffices to have

$$
(\eta + A_\Phi)\frac{\alpha^2 p(1-p)}{G}(1 + B^2) + (B_\Phi - \eta/2) + 3c\delta(\eta + C_\Phi)(2\alpha p B^2 + 2\alpha^2 p(1-p)) \leq -\eta/8.
$$

This condition is equivalent to the following condition:

$$
20\frac{\alpha(1-p)\left(1 + B^2\right)}{G} + 60c\delta\left(B^2 + \alpha(1-p)\right) \leq 1.
$$

Therefore, if $p < 1$ and $\delta < \frac{1}{60cB^2}$, this is equivalent to

$$
\alpha \leq \frac{G\left(1 - 60c\delta B^2\right)}{(1-p)\left(20\left(1 + B^2\right) + 60c\delta G\right)} =: \alpha_G.
$$

If $\alpha = \sqrt{90}L\eta/p$, this implies

$$
\eta \leq \frac{p}{\sqrt{90}L}\alpha_G = \eta_G.
$$

Therefore,

$$\frac{\eta}{8} \sum_{t=1}^{T-1} \mathbb{E}\|\nabla f(\boldsymbol{x}^t)\|^2 \le \Phi_1 - \Phi_T$$
$$+ (T-1)\left[\frac{5\eta}{2}\left(\frac{\alpha\sigma^2}{G} + \frac{\alpha(1-p)\zeta^2}{G}\right) + 3c\delta\left(\frac{45L^2\eta^3}{2\alpha^2p^2} + \eta\right)(2\zeta^2 + 2\alpha\sigma^2)\right]$$

Since $\Phi_1 := \mathbb{E}f(\boldsymbol{x}^1) - f^* + A_\Phi \bar{e}_1 + B_\Phi \mathbb{E}\|\nabla f(\boldsymbol{x}^0)\|^2 + 3C_\Phi c\delta\varrho_1$, by using descent lemma on $\mathbb{E}f(\boldsymbol{x}^1)$,

$$\Phi_1 \le f(\boldsymbol{x}^0) - f^* - (\frac{\eta}{2} - B_\Phi)\mathbb{E}\|\nabla f(\boldsymbol{x}^0)\|^2 + (\eta + A_\Phi)\bar{e}_1 + 3c\delta(\eta + C_\Phi)\varrho_1$$
$$\le f(\boldsymbol{x}^0) - f^* - \frac{\eta}{8}\mathbb{E}\|\nabla f(\boldsymbol{x}^0)\|^2 + \frac{5\eta\sigma^2}{2\alpha pG} + 3c\delta\left(\frac{45L^2\eta^3}{2\alpha^3p^3} + \frac{\eta}{\alpha p}\right)(2\sigma^2 + \zeta^2)$$

Therefore,

$$\frac{\eta}{8}\sum_{t=0}^{T-1}\mathbb{E}\|\nabla f(\boldsymbol{x}^t)\|^2 \le f(\boldsymbol{x}^0) - f^*$$
$$+ (T-1)\left[\frac{5\eta}{2}\left(\frac{\alpha\sigma^2}{G} + \frac{\alpha(1-p)\zeta^2}{G}\right) + 3c\delta\left(\frac{45L^2\eta^3}{2\alpha^2p^2} + \eta\right)(2\zeta^2 + 2\alpha\sigma^2)\right]$$
$$+ \frac{5\eta\sigma^2}{2\alpha pG} + 3c\delta\left(\frac{45L^2\eta^3}{2\alpha^3p^3} + \frac{\eta}{\alpha p}\right)(2\sigma^2 + \zeta^2)$$

Dividing both sides by $\eta/8$ and $T$, we have

$$\frac{1}{T}\sum_{t=0}^{T-1}\mathbb{E}\|\nabla f(\boldsymbol{x}^t)\|^2 \le \frac{8(f(\boldsymbol{x}^0) - f^*)}{\eta T}$$
$$+ 20\left(\frac{\alpha\sigma^2}{G} + \frac{\alpha(1-p)\zeta^2}{G}\right) + 24c\delta\left(\frac{45L^2\eta^2}{2\alpha^2p^2} + 1\right)(2\zeta^2 + 2\alpha\sigma^2)$$
$$+ \frac{20\sigma^2}{\alpha pGT} + \frac{24c\delta}{T}\left(\frac{45L^2\eta^2}{2\alpha^3p^3} + \frac{1}{\alpha p}\right)(2\sigma^2 + \zeta^2).$$

Substituting $\alpha = \sqrt{90}L\eta/p$:

$$\frac{1}{T}\sum_{t=0}^{T-1}\mathbb{E}\|\nabla f(\boldsymbol{x}^t)\|^2 \le \frac{1}{\eta T}\left(8(f(\boldsymbol{x}^0) - f^*) + \frac{20\sigma^2}{\sqrt{90}LG} + \frac{24c\delta(2\sigma^2 + \zeta^2)}{\sqrt{90}L}\right)$$
$$+ \eta\left[\left(\frac{20L\sigma^2}{pG} + \frac{20L(1-p)\zeta^2}{pG}\right) + \frac{60c\delta\sigma^2 L}{p}\right]$$
$$+ 60c\delta\zeta^2$$

Since $\alpha := \min(1, \sqrt{90}L\eta/p)$, This requires the condition on $\eta \le \frac{p}{\sqrt{90}L}$. Also, by the conditions above, we restricted the range of $\eta$ as

$$0 < \eta \le \min\{\frac{p}{\sqrt{90}L}, \eta_G\} = \frac{p}{\sqrt{90}L\max\{1,\Gamma\}},$$

where $\Gamma := \frac{(1-p)(20(1+B^2)+60c\delta G)}{G(1-60c\delta B^2)}$. Invoking the stepsize tuning Lemma E.9, we set $\eta$ as:

$$\eta = \min\left\{\frac{p}{\sqrt{90}L\max\{1,\Gamma\}}, \sqrt{\frac{8(f(\boldsymbol{x}^0) - f^*) + \frac{20\sigma^2}{\sqrt{90}LG} + \frac{24c\delta(2\sigma^2+\zeta^2)}{\sqrt{90}L}}{\left[\left(\frac{20L\sigma^2}{pG} + \frac{20L(1-p)\zeta^2}{pG}\right) + \frac{60c\delta\sigma^2 L}{p}\right]T}}\right\},$$

we have the following convergence rate:

$$\frac{1}{T}\sum_{t=0}^{T-1}\mathbb{E}\|\nabla f(\boldsymbol{x}^t)\|^2 \le 60c\delta\zeta^2 + 2\sqrt{\frac{br_0}{T}} + \frac{dr_0}{T}$$

$$\le \mathcal{O}\left( c\delta\zeta^2 + \sqrt{\frac{(LF^0 + c\delta(\sigma^2 + \zeta^2))(\sigma^2 + (1-p)\zeta^2)}{pGT}} + \frac{\sigma}{G}\sqrt{\frac{\sigma^2 + (1-p)\zeta^2}{pT}} \right.$$

$$\left. + \sqrt{\frac{c\delta\sigma^2\left(LF^0 + \frac{\sigma^2}{G} + c\delta(\sigma^2 + \zeta^2)\right)}{pT}} + \frac{\max\{1,\Gamma\}(LF^0 + \frac{\sigma^2}{G} + c\delta(\sigma^2 + \zeta^2))}{pT} \right)$$

$$\le \mathcal{O}\left( c\delta\zeta^2 + \sqrt{\frac{(LF^0 + c\delta(\sigma^2 + \zeta^2))(\sigma^2 + (1-p)\zeta^2)}{pGT}} + \sqrt{\frac{c\delta\sigma^2\left(LF^0 + \frac{\sigma^2}{G} + c\delta(\sigma^2 + \zeta^2)\right)}{pT}} \right.$$

$$\left. + \frac{\max\{1,\Gamma\}(LF^0 + \frac{\sigma^2}{G} + c\delta(\sigma^2 + \zeta^2))}{pT} \right).$$

$\square$

**Lemma E.9** (Stepsize Tuning). *Let $C \ge 0$, $r_0 > 0$, $\bar{\eta} > 0$, $b \ge 0$, and $T \ge 1$. Assume that for every $\eta \in (0, \bar{\eta}]$,*

$$R_T(\eta) \le C + \frac{r_0}{\eta T} + b\eta.$$

*Then the following hold.*

*(i) If $b > 0$, define*

$$\eta_T^\star := \min\left\{ \bar{\eta}, \sqrt{\frac{r_0}{bT}} \right\}.$$

*Then*

$$R_T(\eta_T^\star) \le C + \begin{cases} 2\sqrt{\dfrac{br_0}{T}}, & \text{if } T \ge \dfrac{r_0}{b\bar{\eta}^2}, \\ \dfrac{r_0}{\bar{\eta}T} + b\bar{\eta}, & \text{if } T < \dfrac{r_0}{b\bar{\eta}^2}. \end{cases}$$

*(ii) In particular, if $b > 0$,*

$$R_T(\eta_T^\star) \le C + 2\sqrt{\frac{br_0}{T}} + \frac{r_0}{\bar{\eta}T}.$$

*(iii) If $b = 0$, the choice $\eta_T^\star = \bar{\eta}$ gives*

$$R_T(\eta_T^\star) \le C + \frac{r_0}{\bar{\eta}T}.$$

*Proof.* Define

$$\phi_T(\eta) := \frac{r_0}{\eta T} + b\eta, \qquad \eta \in (0, \bar{\eta}].$$

Then the assumed bound reads

$$R_T(\eta) \le C + \phi_T(\eta).$$

Assume first that $b > 0$. Since $r_0 > 0$,

$$\phi_T'(\eta) = -\frac{r_0}{\eta^2 T} + b, \qquad \phi_T''(\eta) = \frac{2r_0}{\eta^3 T} > 0 \quad \text{for all } \eta > 0.$$

Hence $\phi_T$ is strictly convex on $(0, \infty)$, and its unique unconstrained minimizer is obtained from $\phi'_T(\eta) = 0$, namely

$$\eta_{\text{unc}} = \sqrt{\frac{r_0}{bT}}.$$

Therefore the minimizer of $\phi_T$ over the interval $(0, \bar{\eta}]$ is exactly

$$\eta_T^{\star} = \min\{\bar{\eta}, \eta_{\text{unc}}\} = \min\left\{\bar{\eta}, \sqrt{\frac{r_0}{bT}}\right\}.$$

If $\eta_{\text{unc}} \leq \bar{\eta}$, equivalently $T \geq r_0/(b\bar{\eta}^2)$, then

$$\phi_T(\eta_T^{\star}) = \frac{r_0}{T\sqrt{r_0/(bT)}} + b\sqrt{\frac{r_0}{bT}} = 2\sqrt{\frac{br_0}{T}}.$$

If $\eta_{\text{unc}} > \bar{\eta}$, equivalently $T < r_0/(b\bar{\eta}^2)$, then $\eta_T^{\star} = \bar{\eta}$, and thus

$$\phi_T(\eta_T^{\star}) = \frac{r_0}{\bar{\eta}T} + b\bar{\eta}.$$

This proves part (i).

For part (ii), in the first regime we simply add the nonnegative term $r_0/(\bar{\eta}T)$. In the second regime, $\bar{\eta} < \sqrt{r_0/(bT)}$, so

$$b\bar{\eta} < \sqrt{\frac{br_0}{T}}.$$

Hence

$$\phi_T(\eta_T^{\star}) = \frac{r_0}{\bar{\eta}T} + b\bar{\eta} < \frac{r_0}{\bar{\eta}T} + \sqrt{\frac{br_0}{T}} \leq \frac{r_0}{\bar{\eta}T} + 2\sqrt{\frac{br_0}{T}}.$$

This proves part (ii).

Finally, if $b = 0$, then

$$\phi_T(\eta) = \frac{r_0}{\eta T}$$

is strictly decreasing on $(0, \bar{\eta}]$, so its minimum over $(0, \bar{\eta}]$ is attained at $\eta = \bar{\eta}$. Therefore

$$R_T(\bar{\eta}) \leq C + \frac{r_0}{\bar{\eta}T}.$$

This proves part (iii). $\qquad\qquad\qquad\qquad\qquad\qquad\qquad\qquad\qquad\qquad\qquad\qquad\qquad\qquad\quad\square$

