# OpenReview forum: "Delayed Momentum Aggregation: Communication-efficient Byzantine-robust Federated Learning with Partial Participation"
_ICML.cc/2026/Conference — ICML 2026 regular_

### Official Review · Reviewer_pAun · 2026-03-03

**Soundness:** 3
**Presentation:** 4
**Significance:** 2
**Originality:** 3
**Overall Recommendation:** 3
**Confidence:** 4

**Summary:**

The paper presents a Byzantine-robust Federated Learning method that supports partial participation. The key concept of Delayed Momentum is used to avoid a Byzantine majority in the selected updates. It relies on using cached momenta of the clients not participating in the current round, to which a preprocessing function is applied to remove the additional variance.  Theoretical convergence to a neighborhood of the local optimum (or saddle point) is established. Experiments on MNIST and CIFAR-10 are provided to illustrate the practicality of the method.

**Compliance With Llm Reviewing Policy:**

Affirmed.

**Final Justification:**

The paper has many positive aspects, key of which is trading off communication for memory to improve robustness which is a neat idea that allows supporting "low" partial participation scenarios that were not covered before in the literature. However, although the authors rebuttal did answer some of my concerns, it remains unclear whether the chosen attacks are strong enough and whether the theoretical assumptions are practical. These are both important points in my view and I will therefore maintain my original score.

**Key Questions For Authors:**

1. Do the authors compare with (Allouah et al. 2024) in general, and in particular in situations where no Byzantine majority is ever encountered (I believe that Allouah et al. (2024) presents conditions for which Byzantine majority is unlikely to occur) ?

**Limitations:**

yes

**Strengths And Weaknesses:**

**Strengths**

- The presented method is a simple yet conceptually powerful way to prevent Byzantine-majority rounds. The method does not incur higher communication costs than previous methods.

- The authors provide convergence guarantees and discuss how these compare with the literature.

- The presentation is clear and the algorithm design is well explained.

**Weaknesses**

1. The theoretical assumptions are more constraining than the authors claim. Although up to constants the condition on $\delta$ seems on par with the $1/(2+ B^2)$ in the literature, the breakdown point found in this paper is much smaller reducing the guarantees to less than 1% byzantines in practice. The condition on $G$ is not well justified and it is not clear to me what the authors mean by "$B^2$ can be made small by adjusting the parameterization".

2. While the algorithm does not result in a higher communication cost, the memory cost is expensive, being that of full participation which can be inhibitory for practical settings.

3. Contrary to the previous work on partial participation (Allouah et al 2024), multiple local updates are not supported.

4. The experiments are quite limited :
    - The size of client pool is very small, and the chosen value of $p=0.1$ does not seem to make a lot of sense when $n=10$.
    - In Figure 2, the training loss for non-iid seems to go down much faster than the iid setting which is a little suspicious.
    - Figures 7 and 8 show that the algorithm fails with KRUM and CM, both in IID and non-IID settings.
    - Some robust aggregation methods such as Coordinate-wise Trimmed mean and pre-aggregations are not included.
    - In Figure 6, the convergence of the plain average despite $\delta=0.2$ is another cause of suspicion that indicates that the attacks are not strong, or were not implemented and optimized properly.

5. The paper does not discuss the effect of obsolete momenta and how they can be exploited by attackers. These effects would be more present in low sampling rate scenarios.

**A few small remarks**
- Definition 2.4 should be referenced with the original citation from Karimireddy et al. (2021) instead of the subsequent work of Gorbunov et al. (2023)
- Typo in line 370 (right) “while FedCM …”
- The most successful aggregation, CCLIP, was not defined in the main paper, and can be hardly found in the appendix.

---

> ### Author Rebuttal · Authors · 2026-03-31
>
> We thank the reviewer for the constructive feedback. We have addressed your concerns about the additional experiments and conditions in Theorem 3.1, including the comparison with (Allouah et al., 2024). We kindly ask the reviewer to reconsider your evaluation and, if possible, adjust the score to reflect these changes.
>
> ---
>
> # Key Question: Detailed Comparison with (Allouah et al., 2024)
> We conducted additional experiments under fixed-size client sampling, reflecting the setting in Allouah et al. (2024) where Byzantine majority cannot occur by construction.
>
> ## Experimental Setup
> - Total clients: n = 10, subsample size: s = 5, Byzantine clients: b = 2 ( $\delta = 0.2$ ), seeds: 0,1.
> - The DeMoA preprocessing function is adapted from the Bernoulli case by $p = s/n$.
> - Aggregation rule: Centered Clipping (CCLIP), as they performed best in our main experiments.
> - All supplementary figures are available at the anonymous repository: https://anonymous.4open.science/r/rebuttal-materials-for-icml2026-3472 (hereafter **[Repo]**).
> - Results: **[Repo]/fixedsize/**  (PDF files for MNIST and CIFAR-10 experiments). The red curve (FedAvg) corresponds to the optimizer of Allouah et al. (2024).
>
> ### Byzantine CIFAR-10 ResNet (Fixed-Size, n=10, b=2, s=5), CCLIP (Non-IID)
> | Optimizer | ALIE | BF | IPM | LF | Mimic |
> |---|---|---|---|---|---|
> | FedAvg | 48.00 ± 2.24 | 10.00 ± 0.00 | 47.02 ± 0.97 | 45.60 ± 1.47 | 44.37 ± 2.16 |
> | FedCM | 47.81 ± 2.61 | 10.00 ± 0.00 | 47.35 ± 1.35 | 45.27 ± 1.35 | 45.24 ± 1.29 |
> | DeMoA | **65.21 ± 0.38** | 10.00 ± 0.00 | **65.43 ± 0.20** | **63.70 ± 0.34** | **62.69 ± 1.74** |
> | Byz-VR-MARINA-PP | 23.27 ± 0.71 | **11.68 ± 1.68** | 14.32 ± 3.08 | 24.96 ± 0.81 | 18.07 ± 3.42 |
>
>
> Under fixed-size sampling (no Byzantine majority), FedAvg performs comparably to DeMoA, consistent with Allouah et al. (2024). However, as shown in the table above, DeMoA outperforms all baselines under most attacks. Further, we would like to emphasize the critical advantage of DeMoA lies in its **generality**: it maintains robustness even when Byzantine majority rounds can occur, whereas FedAvg and FedAvgM collapse in such rounds.
>
> ---
> # Weaknesses
> We thank the reviewer for raising concerns. We address each of them in turn.
> ## On Assumptions and Conditions for Theorem 3.1
> ### On Breakdown Point
> Due to the character limit, we refer to our detailed response to reviewer vLwB on DeMoA's viability for the full discussion.
>
> ### Condition for $(\zeta, B)$-Heterogeneity
>
> The parametrization for $(\zeta, B)$-heterogeneity is not unique: one can choose a larger $\zeta$ to obtain a smaller $B$ (at the cost of a larger non-vanishing term). This is precisely what "$B$ can be made small" refers to: the practitioner can tune this trade-off to relax the condition on $\delta$.
>
> ---
>
> ## On Memory Cost, Multiple Local Updates, and Scale of Experiments
> Due to the character limit, we refer to our detailed responses to reviewers vLwB (memory cost, multiple local updates) and jSwB (scale of experiments). We promise to add large-scale experiments in the camera-ready version.
>
> ## On Figure 2 (Non-IID vs. IID Training Loss)
> This was caused by compressing the axis to save space in the manuscript. We have updated the figures with consistent axis scaling: see [Repo]/figure2.pdf, figure4a_right.pdf, figure4b_left.pdf, figure4b_right.pdf.
>
> ## On Figures 7 and 8 (Failure with KRUM and CM)
> These figures intentionally illustrate that DeMoA may fail under certain aggregator combinations. This does not reflect a fundamental limitation of DeMoA itself, but rather highlights the importance of choosing an aggregator that is compatible with the theoretical conditions outlined in our convergence analysis.
>
> ## On Missing (Pre-)Aggregators
> Pre-aggregation (bucketing) is indeed included in the non-IID experiments.
> **Additionally, we conducted experiments with ARC [1], NNM (Allouah et al., 2023a), and Trimmed Mean (TrMean) on the [Repo]**. We will add details of the experiments' setup in the revised version.
>
> ## On Suspicion for Naive Average
> Our implementation is based on the publicly available codebase (URL in paper), so the attack implementations follow established standards. The plain average is indeed vulnerable to stronger attacks (e.g., $\|m_i\| \to \infty$), which were not included in our benchmark, explaining the observed convergence.
>
> ## On Attacks against Leveraging Delayed Momentum
> Due to the character limit, we refer to our detailed response to reviewer zN6L on adaptive attacks.
>
> ---
>
> # Minor Remarks
> We thank the reviewer for these careful observations. We will fix all three in the camera-ready version. We note that our theoretical analysis adopts the generalized definition from Gorbunov et al. (2023), which we will make explicit.
>
> ---
>
> [1] Y. Allouah, R. Guerraoui, N. Gupta, A. Jellouli, G. Rizk, and J. Stephan. Adaptive Gradient Clipping for Robust Federated Learning, In The Thirteenth International Conference on Learning Representations, 2024.

---

> > ### Author Rebuttal · Reviewer_pAun · 2026-04-03
> >
> > I thank the authors for the rebuttal. I appreciate the different details, however, I believe several points remain unsolved.
> > - The plain average having almost perfect accuracy scores for all chosen attacks might indicate that something is wrong with the implementation of the attacks. The different attacks used are enough to break the plain average especially on CIFAR-10. Attack such as ALIE also have tunable parameters that should be well chosen.
> > - The answer to Reviewer vLwB states that the memory issue is orthogonal, but it is not. In the cross-device scenario, partial participation is not only due to clients occasionally disconnecting from the training loop, but also due to the extremely high communication cost and impracticality of total participation.
> > - The theoretical condition on $G$ remains quite restrictive and should be discussed more. It is a bit odd to have a condition on the absolute number of honest clients, even when the Byzantine fraction is $0$.
> > - The answer to Reviewer zN6L on adaptive attacks is insufficient. Although the theory does show that arbitrary attacks can be supported (under some conditions), the goal of the experiments is to show that in practice the method is robust, even if the theoretical conditions may not be totally satisfied (for instance the condition on $G$ is not satisfied for any of the experiments). It is therefore necessary to at least make a best effort adaptation of the previous attacks (which might have been the case, but it is not clear from the paper).

---

> > > ### Author Response · Authors · 2026-04-08
> > >
> > > # Correction in Experimental Configuration (Conclusions Unchanged)
> > >
> > > We thank the reviewers for their careful feedback, which prompted us to re-inspect our code and discover a configuration issue in the sampling routine, which primarily affects the low participation regime with a small number of clients, allowed DeMoA to benefit from more client updates than intended.
> > >
> > > We have corrected this and re-run experiments. **DeMoA continues to outperform all existing methods in most cases, and our conclusions still hold.** Corrected results for CIFAR-10 and MNIST are provided below and the figures are under **[Repo/updated_figures]**:
> > >
> > > - Baseline CIFAR-10 ($p=0.5, n=20$) [figure2_p0.5.pdf]:DeMoA remains superior, including with non-Byzantine clients.
> > > - Byzantine MNIST ($p=0.5, n=25$) [figure1_a_p0.5.pdf]: results closely match the original submission, with improvements on certain attacks (e.g., ALIE).
> > > - Byzantine MNIST ($p=0.1, n=25$) [figure1_a_p0.1.pdf]: DeMoA succeeds even in the low participation regime.
> > > - Byzantine CIFAR-10 ($p=0.5, n=10$) [figure1_b_p0.5.pdf]: DeMoA succeeds in large-scale experiments with moderate participation rate.
> > >
> > > We note that for $p=0.1$ on CIFAR-10 without Byzantine clients, the effective training budget is only 20 epochs in expectation, which is too few for any method, including FedAvg and FedAvgM, to train ResNet-10 (see [figure2_p0.1.pdf]).
> > > We promise to include experiments with sufficient epochs and a larger client pool for this setting in the camera-ready version.
> > > These larger-scale experiments requires more computational time which was unavailable within the rebuttal period.
> > >
> > > We are grateful to the reviewers for motivating this closer inspection and for strengthening the rigor of our evaluation.
> > >
> > > # On Plain Average Experiments
> > >
> > > We appreciate the reviewer for indicating this point. We humbly respond that the near-perfect accuracy of plain average is not due to an implementation error, but rather a natural consequence of the interplay between client sampling and the Byzantine ratio in our setting.
> > >
> > > In our configuration ($n=25,f=5,\delta=0.2,p=0.1$), the probability that a sampled subset contains zero Byzantine clients is $(1-p)^f = 0.9^5 \approx 0.6$. That is, in the majority of rounds, the plain average operates over an entirely honest sample. Over a long time horizon $T$, these Byzantine-free rounds dominate, allowing the plain average to appear robust. We note that extreme attacks can break plain average with even a single malicious worker in a single round, but the non-extreme attacks shown in our figures are naturally mitigated by this sampling effect.
> > >
> > > Furthermore, we agree that ALIE's tunable parameter $\tau$ is important. However, increasing $\tau$ to strengthen the attack simultaneously makes the malicious updates more distinguishable from honest ones, which undermines ALIE's core design principle of remaining indistinguishable.
> > >
> > > # On Memory Issue
> > >
> > > We thank the reviewer for pressing on this important point. After careful consideration, we agree that communication efficiency is a critical consideration in cross-device settings and not merely orthogonal.
> > >
> > > We humbly note, however, that this challenge remains open across the Byzantine-robust FL with PP literature. For instance, Allouah et al. (2024) do not address  compression. Malinovsky et al. (2024) do incorporate compression, but their approach leverages a stochastic gradient structure that is naturally amenable to it; their provable guarantees apply to their MARINA-based optimizer, which does not extend to deep learning settings, and their heuristic extension for deep learning tasks lacks formal guarantees.
> > > We elaborate further on the technical difficulties of combining momentum-based methods with compression in our response to Reviewer jSwB.
> > >
> > > To our knowledge, no existing Byzantine-robust FL method with PP fully resolves the communication efficiency problem with provable guarantees that works for deep learning. We agree with the reviewer that this as an important direction for future work and will discuss it explicitly in the camera-ready version.
> > >
> > > # On Dependence on $G$
> > >
> > > We thank the reviewer for this observation. We are pleased to report that we have successfully removed the dependence on $G$ entirely. Specifically, we replace the condition on $G$ (line 1280) with a condition on $\alpha$, and then substitute $\alpha = \sqrt{90}\, L\eta/p$ to convert it into a condition on the global step size $\eta_G$. We then define $\eta_{\text{new}} := \min (\eta, \eta_G)$.
> > >
> > > This only introduces an additional term in the $\mathcal{O}(1/T)$ component of the convergence bound, leaving the dominant $\mathcal{O}(1/\sqrt{T})$ term unchanged. We will include the updated proof, revised theorem statements, and corresponding discussion in the camera-ready version.
> > >
> > > # On Adaptive Attacks
> > >
> > > We thank the reviewer for pressing on this point.
> > > Due to character limit, we kindly ask the reviewer to refer to our latest response to Reviewer zN6L.

---

### Official Review · Reviewer_vLwB · 2026-03-04

**Soundness:** 2
**Presentation:** 3
**Significance:** 2
**Originality:** 2
**Overall Recommendation:** 4
**Confidence:** 4

**Summary:**

In federated learning scenarios with partial client participation, a large proportion of sampled clients may be Byzantine, thereby severely degrading model performance. This paper proposes a delayed momentum aggregation mechanism that allows the central server to combine cached momentum from unsampled clients with fresh momentum from sampled clients, effectively mitigating the influence of excessive Byzantine clients. The authors integrate this mechanism into the DeMoA optimizer and analyze its convergence properties. Moreover, they demonstrate the effectiveness of the proposed algorithm through extensive experiments.

**Compliance With Llm Reviewing Policy:**

Affirmed.

**Key Questions For Authors:**

1)	In federated learning with partial client participation, the total number of clients can be extremely large, raising concerns about whether the server can feasibly store model updates from all clients simultaneously.
2)	The authors employed a preprocessing function to reduce the variance introduced by delayed momentum. However, neither the description of Algorithm 1 nor the derivations provided in Appendix E reflect the incorporation of this preprocessing function.
3)	In federated learning, clients typically perform multiple local training iterations to reduce communication frequency. However, DeMoA omits this design, raising the question of whether this omission compromises training efficiency.
4)	The authors simulated no more than 20 clients in their experiments, which is a relatively small scale and raises concerns regarding the algorithm’s scalability.
5)	“This behavior arises when the combination of partial participation rate and Byzantine ratio exceeds the breakdown point of the corresponding aggregator.” This phenomenon raises questions about whether DeMoA operates only under specific conditions. The authors also do not discuss the reasons for DeMoA's poor performance. The authors should further analyze and clarify the conditions required for the algorithm’s viability.

**Limitations:**

The authors should further discuss the feasibility of the proposed algorithm in large-scale federated learning systems and clarify the role of the preprocessing function in both the algorithmic procedure and the theoretical analysis.

**Strengths And Weaknesses:**

Strengths
1)	The algorithm proposed by the authors effectively addresses the challenge posed by an excessive number of Byzantine clients in scenarios with partial client participation, thereby enhancing the robustness of model training.
2)	DeMoA reduces the variance introduced by delayed momentum and theoretically guarantees the convergence of trained models.

Weakness
1)	The DeMoA algorithm incurs substantial storage overhead, but the authors do not provide a detailed discussion of this issue.
2)	The role of the preprocessing function in the paper is questionable.
3)	The authors did not provide an adequate discussion of the algorithm’s robustness.

---

> ### Author Rebuttal · Authors · 2026-03-31
>
> We thank the reviewer for the feedback. Below, we provide responses to each of the points raised. We hope the below points are helpful and kindly ask the reviewer to consider them when finalizing the score.
>
> ## Key Questions and Weaknesses:
> ### On Memory Cost
> > In federated learning with partial client participation, the total number of clients can be extremely large, raising concerns about whether the server can feasibly store model updates from all clients simultaneously.
>
> We thank the reviewer for the comment. While the current design does incur memory costs proportional to the total number of clients, we wish to make two clarifying remarks.
>
> First, our primary contribution is to establish that Byzantine-robust FL with partial participation is achievable via a simple, principled solution. Memory efficiency is an orthogonal concern that we intentionally defer.
>
> Second, DeMoA is naturally compatible with existing memory-reduction techniques.
> For example, gradient compression (e.g., EF-based compressors [1]) can possibly be incorporated directly into the preprocessing function $\mathcal{P}$, reducing the server-side memory footprint.
>
> We therefore view memory compression as a promising and tractable direction for future work, rather than a limitation intrinsic to DeMoA's design.
>
> ### Description of Preprocessing
> > However, neither the description of Algorithm 1 nor the derivations provided in Appendix E reflect the incorporation of this preprocessing function.
>
> We acknowledge that the connection between Algorithm 1 and the preprocessing function was not sufficiently emphasized in the main text. We kindly refer the reviewer to **Appendix A**, where we provide a detailed motivation, including a comparison with related methods and a discussion of the design choices behind its formulation. We will make this connection more explicit in the revised manuscript.
>
> ### Multiple Local Training
> > However, DeMoA omits this design, raising the question of whether this omission compromises training efficiency.
>
> We thank the reviewer for pointing out this limitation. We note that supporting multiple local updates remains an open challenge more broadly, it is not addressed in the closest related work (Malinovsky et al., 2024), nor in several prominent Byzantine-robust FL methods even under full participation (Karimireddy et al., 2021, 2022; Gorbunov et al., 2023). To our knowledge, (Allouah et al., 2024) is the only work to handle this in the partial participation setting.
> Our primary contribution is establishing the feasibility of Byzantine-robust FL with partial participation in the simplest possible form. Adding multiple local updates would obscure this core message. However, we consider extending DeMoA to support multiple local updates an important direction for future work.
>
>
> ### Scale of Experiments
> > The authors simulated no more than 20 clients in their experiments, which is a relatively small scale and raises concerns regarding the algorithm’s scalability.
>
> We thank the reviewer for the suggestion. Due to the short period in this rebuttal, we cannot finish the experiments. We promise to add large-scale CIFAR-10 experiments, e.g., with $n=25$, in the camera-ready version.
>
> ### Concerns of DeMoA's Viability
> > “This behavior arises when the combination of partial participation rate and Byzantine ratio exceeds the breakdown point of the corresponding aggregator.” This phenomenon raises questions about whether DeMoA operates only under specific conditions. The authors also do not discuss the reasons for DeMoA's poor performance. The authors should further analyze and clarify the conditions required for the algorithm’s viability.
>
> We thank the reviewer for prompting clarification.
> The condition for DeMoA's viability is the breakdown point condition in our convergence theorem, depending on aggregator constant $c$, partial participation ratio $p$, and momentum parameter $\alpha$. The failure cases in Figures 7 and 8 arise when this condition is violated, a phenomenon also observed in non-i.i.d. bucketing (Karimireddy et al., 2022).
>
> We respectfully indicate that our breakdown point may not be more restrictive than the literature. Specifically, if the time horizon $T$ is sufficiently large (which is typical for deep learning training), one can choose a very small momentum $\alpha$, so that the effective condition on $\delta$ asymptotically reduces to $\min\left(\frac{1}{2}, \frac{1}{60cB^2}\right) = \mathcal{O}\left(\frac{1}{2+B^2}\right),$  which **matches the known upper bound for the breakdown point** in Byzantine-robust FL (Allouah et al., 2023b), even when $p < 1$. Moreover, in the vast majority of our experiments, DeMoA performs on par with or better than existing methods, suggesting this condition is mild in practice.
>
> ---
>
> [1] Fatkhullin, I., Tyurin, A., & Richtárik, P. Momentum provably improves error feedback!. In Advances in Neural Information Processing Systems, 2023.

---

> > ### Author Rebuttal · Reviewer_vLwB · 2026-04-03
> >
> > The rebuttal solved my concerns, so I maintain my score.

---

> > > ### Author Response · Authors · 2026-04-08
> > >
> > > We sincerely thank the reviewer for acknowledging our previous responses. We would also like to note that during this rebuttal process, we identified and corrected a configuration issue in our experiments. The corrected results confirm that our conclusions remain unchanged, as detailed in **Section: Correction in Experimental Configuration** in our response to Reviewer pAun.

---

### Official Review · Reviewer_zN6L · 2026-03-12

**Soundness:** 3
**Presentation:** 3
**Significance:** 3
**Originality:** 3
**Overall Recommendation:** 4
**Confidence:** 4

**Summary:**

This paper addresses Byzantine-robust federated learning under partial participation, a practical but mostly underexplored setting. The authors propose DeMoA, which aggregates fresh momentum from sampled clients and cached momentum from inactive ones. This maintains a server-side honest majority without extra communication costs, offering both theoretical guarantees and good empirical performance.

**Compliance With Llm Reviewing Policy:**

Affirmed.

**Final Justification:**

I still have some concerns regarding W2, and  I remain somewhat concerned about the robustness under adaptive attacks. Thus I maintain my original score.

**Key Questions For Authors:**

The majority of the concerns are outlined in the 'Weaknesses' section.

Moreover, I think there are some related papers who also use the  historical information to improve model performance in FL such as

[1] Jhunjhunwala, D., Sharma, P., Nagarkatti, A., & Joshi, G. Fedvarp: Tackling the variance due to partial client participation in federated learning.

[2] Wang, Y., Cao, Y., Wu, J., Chen, R., & Chen, J.. Tackling the data heterogeneity in asynchronous federated learning with cached update calibration.

**Limitations:**

Yes.

**Strengths And Weaknesses:**

Strengths:

* The proposed method is well-motivated and provides some non-trivial insight into the field, the overall paper is well-written and easy to follow.
* The delayed momentum aggregation principle is conceptually simple and easy to understand.
* The convergence analysis is thorough and is accompanied by detailed discussion and illustrations.
* The experimental results are convincing.

Weaknesses:
* I think the primary weakness of this paper is the insufficient evaluation of adaptive attacks. Current experiments primarily target classical predefined attacks. However, if attackers know the sampling mechanism of an algorithm, they may devise attack strategies specifically targeting “delayed momentum.” For instance, they could exploit the lag in historical information through carefully designed gradient drift. I recommend the authors include discussions or experiments addressing this concern.
* Quite related to the first point, the algorithm's correction coefficient is highly dependent on $p_t$. In practical applications, client participation is typically unpredictable. If the server's estimate of the participation rate is biased, will the algorithm's performance degrade?
* DeMoA requires the server to store the momentum vectors of all clients. When the number of clients $n$ scales to thousands or larger, the server's memory pressure becomes extremely significant. Have the authors considered memory reduction techniques?

---

> ### Author Rebuttal · Authors · 2026-03-31
>
> We thank the reviewer for constructive comments. Below, we have addressed the reviewer's concerns in turn. We hope our clarifications address the concerns and that they will be reflected in the final score.
>
> ## Key Questions
> ### Additional References
> We thank the reviewer for pointing out these related works. While both [1] and [2] leverage historical information to address partial participation and data heterogeneity, they differ from our approach in that they do not incorporate robust aggregators or momentum-based variance reduction. We will include them in the related work section of the camera-ready version.
>
> ## Weaknesses
> ### On Adaptive Attacks
> > I think the primary weakness of this paper is the insufficient evaluation of adaptive attacks. Current experiments primarily target classical predefined attacks. However, if attackers know the sampling mechanism of an algorithm, they may devise attack strategies specifically targeting “delayed momentum.” For instance, they could exploit the lag in historical information through carefully designed gradient drift. I recommend the authors include discussions or experiments addressing this concern.
>
> We thank the reviewer for this constructive comment. We note that our convergence guarantee is derived under worst-case scenario, which includes adaptive attacks targeting the delayed momentum principle. The robustness (i.e., convergence rate) of DeMoA holds regardless of how Byzantine clients craft their updates, and this is precisely the guarantee our theoretical analysis intends to provide.
>
> ### On Client Availability
> > Quite related to the first point, the algorithm's correction coefficient is highly dependent on $p_t$. In practical applications, client participation is typically unpredictable. If the server's estimate of the participation rate is biased, will the algorithm's performance degrade?
>
> We thank the reviewer for this question. In FL with partial participation, a known or fixed participation rate is a standard assumption in the literature (McMahan et al., 2017; Kairouz et al., 2021). Extending DeMoA to arbitrary or unpredictable participation patterns, as explored in vanilla FL with similar update rule (Gu et al., 2021) to our method, is an interesting direction we leave for future work.
>
> ### On Memory Cost
> > DeMoA requires the server to store the momentum vectors of all clients. When the number of clients $n$ scales to thousands or larger, the server's memory pressure becomes extremely significant. Have the authors considered memory reduction techniques?
>
> We refer to our response to reviewer vLwB for the full discussion. In short, we acknowledge this limitation and view memory-efficient extensions (e.g., via gradient compression in $\mathcal{P}$) as an important direction for future work.

---

> > ### Author Rebuttal · Reviewer_zN6L · 2026-04-03
> >
> > Thank you for the detailed response. However, I still have some concerns regarding W2. Could the authors provide additional experimental results to address this point? Additionally, I remain somewhat concerned about the robustness under adaptive attacks.

---

> > > ### Author Response · Authors · 2026-04-08
> > >
> > > We thank the reviewer for the additional feedback.
> > >
> > > # On More Experimental Results
> > > >  However, I still have some concerns regarding W2. Could the authors provide additional experimental results to address this point?
> > >
> > > We thank the reviewer for the follow-up. Due to the computational demands of these experiments, we were unable to complete them within the rebuttal period. We promise to include these results in the camera-ready version.
> > >
> > > # On Adaptive Attacks
> > > >  Additionally, I remain somewhat concerned about the robustness under adaptive attacks.
> > >
> > > We sincerely thank the reviewer for concern about the robustness under adaptive attacks. This was also pointed out by reviewer pAun, and we understand now that it is an important challenge to address.
> > >
> > > Our experiments do include the mimic attack, which we considered a natural adaptive strategy against DeMoA. However, we acknowledge that we did not fully explore stronger variants that specifically exploit the delayed momentum principle. As the reviewers suggest, an attacker aware of the sampling mechanism could craft an attack to exploit the lag in historical information. Our mimic experiments partially capture this direction, but a dedicated attack targeting the momentum buffers remains unexplored.
> > >
> > > We view this as a valuable research direction and thank the reviewers for highlighting it. We will include a discussion of this in the camera-ready version.
> > >
> > >
> > > ---
> > >
> > > In addition to our responses, we kindly ask the reviewer to refer to the **Section: Correction in Experimental Configuration (Conclusions Unchanged)**  in our rebuttal for the reviewer pAun.
> > > This section includes corrected results after the constructive comments raised by all the reviewers in the previous rebuttal period.

---

### Official Review · Reviewer_jSwB · 2026-03-13

**Soundness:** 3
**Presentation:** 2
**Significance:** 2
**Originality:** 3
**Overall Recommendation:** 4
**Confidence:** 5

**Summary:**

The paper studies Byzantine-robust federated learning under partial participation. The authors point out that when only a subset of clients is sampled in each round, the sampled set may contain a Byzantine majority even if the global Byzantine ratio is below 1/2, which breaks existing robust aggregation methods. To address this issue, they propose a delayed momentum aggregation scheme that combines fresh updates from sampled clients with cached momentum from previously participating clients, and instantiate it in an optimizer called DeMoA. The paper provides convergence analysis under standard assumptions and empirical results on MNIST and CIFAR-10 showing improved robustness compared with several baselines.

**Compliance With Llm Reviewing Policy:**

Affirmed.

**Final Justification:**

The exp is not shown. I maintain my score.

**Key Questions For Authors:**

1. The experiments are conducted with a relatively small number of clients (10--25). Do the authors expect the proposed method to scale to more realistic federated settings with hundreds or thousands of clients? It would be helpful to understand how the performance and system overhead behave as the number of clients increases.
2. The proposed method requires the server to maintain a momentum vector for each client. Could the authors discuss the memory overhead of this design in large-scale FL systems and whether any mechanisms (e.g., compression or pruning of stale states) could mitigate this cost?
3. From Algorithm~1 (line~6), the update suggests a single stochastic gradient step $\nabla f_i(x^{t-1}, \xi_i^t)$ per round. However, the experiments involve deep models (e.g., ResNet-18), where training is typically performed with mini-batches and often multiple local updates. Could the authors clarify how the local training procedure is implemented in practice and how it relates to the algorithm description?
4. The main motivation of the paper is to address Byzantine majority rounds under partial participation. However, the experiments appear to rely on random sampling with a fixed Byzantine ratio, which may only produce such rounds probabilistically. Would it be possible to include controlled experiments where the sampled clients are intentionally dominated by Byzantine clients to directly validate the claimed robustness?

**Limitations:**

The proposed method relies on maintaining historical momentum from all clients to mitigate Byzantine majority rounds. While effective in the evaluated settings, the scalability and robustness of this design in large-scale federated systems with many clients and irregular participation patterns remain to be further validated.

**Strengths And Weaknesses:**

Strengths：
1. The paper focuses on Byzantine-robust federated learning under partial participation, which is a realistic setting but less studied compared to the full participation case.
2. The proposed delayed momentum aggregation is conceptually simple and easy to integrate with existing robust aggregation methods.
3. The paper includes convergence analysis under standard assumptions and empirical results demonstrating improved robustness compared with several baselines.

Weaknesses：
1. The experiments are conducted with a relatively small number of clients (10--25) and on relatively small datasets (MNIST and CIFAR-10). These settings may not fully reflect realistic federated learning scenarios with hundreds or thousands of clients and larger datasets. Evaluations on additional datasets (e.g., SVHN or larger-scale benchmarks such as ImageNet) and larger client populations would help better demonstrate the robustness and scalability of the proposed method.
2. The method requires the server to maintain a momentum vector for each client. While communication overhead is unchanged, this may introduce non-trivial memory overhead when the number of clients is very large. In addition, the method relies on cached momentum from previously participating clients, but the impact of highly stale updates (e.g., when a client has not participated for many rounds) is not clearly analyzed.
3.	From Algorithm~1 (line~6), each sampled client draws a single sample $\xi_i^t \sim D_i$ and computes $\nabla f_i(x^{t-1}, \xi_i^t)$, which suggests a single stochastic gradient step. However, in the experimental setup with deep models (e.g., ResNet-18), training is typically performed using mini-batches and often multiple local updates in realistic federated learning settings. As written, it is unclear how the momentum $m_i^t$ is actually computed in practice and whether multiple mini-batch updates are involved on each client. If I have misunderstood the intended implementation, please correct me, and I would be happy to adjust my review score accordingly.
4.	The theoretical analysis relies on the $(\delta,c)$-robust aggregator assumption, which appears to be a relatively strong requirement. It would be helpful for the authors to clarify whether this assumption is commonly satisfied by widely used Byzantine-robust aggregators such as Krum, coordinate-wise median, or RFA.
5.	A key motivation of the paper is to address the issue of Byzantine majority rounds under partial participation. However, the experimental setup mainly considers a fixed Byzantine ratio with random client sampling, which only leads to Byzantine majority rounds probabilistically. It would strengthen the empirical validation if the authors explicitly constructed scenarios where the sampled clients contain a Byzantine majority, in order to directly demonstrate the robustness of the proposed method under the claimed setting.

---

> ### Author Rebuttal · Authors · 2026-03-31
>
> We thank the reviewer for insightful feedback. Below, we clarify the reviewer's concerns in turn. We hope that these clarifications and revisions can be taken into account when determining the final score.
>
> ## Key Questions and Weakness
>
> ### On Scale of Experiments
> > The experiments are conducted with a relatively small number of clients (10--25). Do the authors expect the proposed method to scale to more realistic federated settings with hundreds or thousands of clients? It would be helpful to understand how the performance and system overhead behave as the number of clients increases.
>
> We thank the reviewer for the suggestion. Due to the short period in this rebuttal, we cannot finish the experiments. We promise to add large-scale CIFAR-10 experiments, e.g., with $n=25$, in the camera-ready version.
> While our experiments are limited to $n = 10,25$ due to computational constraints, our theoretical analysis scales to arbitrarily large $n$. In fact, convergence improves with more clients, as the bound contains a $\mathcal{O}(\sigma^2/\sqrt{GT})$ term where $G$ grows with the number of participating honest clients per round.
>
> ### On the Memory Cost
> > The proposed method requires the server to maintain a momentum vector for each client. Could the authors discuss the memory overhead of this design in large-scale FL systems and whether any mechanisms (e.g., compression or pruning of stale states) could mitigate this cost?
>
> We thank the reviewer for the question. We refer to our response to reviewer vLwB for the full discussion. In short, we acknowledge this limitation, memory-efficient extensions such as gradient compression in $\mathcal{P}$ represent an important direction for future work.
>
> ### On Clarification for Algorithm 1
> > From Algorithm1 (line6), the update suggests a single stochastic gradient step $\nabla f_i(x^{t-1}, \xi_i^t)$
>  per round. However, the experiments involve deep models (e.g., ResNet-18), where training is typically performed with mini-batches and often multiple local updates. Could the authors clarify how the local training procedure is implemented in practice and how it relates to the algorithm description?
>
> We thank the reviewer for asking for clarification.
> As reported in Tables 3 and 4, we use a mini-batch size of 32 to compute stochastic gradients, corresponding to line 6 of Algorithm 1. The stochasticity in $\xi_i^t$ captures this mini-batch sampling in the mathematical formalization. Each client computes a single gradient step per round, which is then sent to the server for robust aggregation.
>
> ### On Byzantine ratio in subsample
> > The main motivation of the paper is to address Byzantine majority rounds under partial participation. However, the experiments appear to rely on random sampling with a fixed Byzantine ratio, which may only produce such rounds probabilistically. Would it be possible to include controlled experiments where the sampled clients are intentionally dominated by Byzantine clients to directly validate the claimed robustness?
>
> This is precisely the setting our experiments evaluate. While the global Byzantine ratio is fixed at $\delta = 0.2$, the Bernoulli sampling mechanism means individual rounds can have a Byzantine-majority subsample, e.g., 3 out of 5 sampled clients being Byzantine. Such rounds are naturally included in our experiments, and DeMoA maintains robustness throughout.
>
>
> ### On Definition of $(\delta,c)$-robust aggregator
> > The theoretical analysis relies on the $(\delta,c)$-robust aggregator assumption, which appears to be a relatively strong requirement. It would be helpful for the authors to clarify whether this assumption is commonly satisfied by widely used Byzantine-robust aggregators such as Krum, coordinate-wise median, or RFA.
>
> We thank the reviewer for the comment.
> This is a standard and widely adopted assumption in the Byzantine-robust FL literature (Karimireddy et al., 2021, 2022; Gorbunov et al., 2023; Malinovsky et al., 2024), with closely related variants used in Allouah et al. (2023) and Farhadkhani et al. (2022).
> The aggregators used in our experiments are known to satisfy this definition, as shown in these prior works.

---

> > ### Author Rebuttal · Reviewer_jSwB · 2026-04-03
> >
> > I am partially OK with the authors' response.
> >
> > I still would like to see the experimental results raised by Q1 and Q4.
> > Regarding Q2, a related experiment is also welcome.
> >
> > I maintain my score.

---

> > > ### Author Response · Authors · 2026-04-08
> > >
> > > We thank the reviewer for additional comments.
> > >
> > > # On Communication Compression
> > > > I still would like to see the experimental results raised by Q1 and Q4. Regarding Q2, a related experiment is also welcome.
> > >
> > > We thank the reviewer for the continued interest in these directions. Regarding communication compression, we would like to highlight that this remains an open challenge across the Byzantine-robust FL literature, not a limitation specific to DeMoA.
> > > The core difficulty lies in the incompatibility of standard compression techniques with Byzantine robustness: Byzantine clients can exploit control variates, which is the key component of communication efficiency, making them unreliable in Byzantine settings. To our knowledge, the only prior work incorporating compression in Byzantine-robust FL with partial participation is Malinovsky et al. (2024), who leverage MARINA-based optimizers.
> > > However, these do not perform well in deep learning settings (Defazio & Bottou, 2019).
> > > Notably, even Malinovsky et al. (2024) do not use compression in their momentum-based extension, which is the variant we used in our experiments.
> > > Addressing this requires new techniques beyond existing tools, and we view it as a promising direction for future work. We will include a discussion of these challenges in the camera-ready version.
> > >
> > > ---
> > >
> > > In addition to our responses, we kindly ask the reviewer to refer to the **Section: Correction in Experimental Configuration (Conclusions Unchanged)**  in our rebuttal for the reviewer pAun.
> > > This section includes corrected results after the constructive comments raised by all the reviewers in the previous rebuttal period.

---

### Decision · Program_Chairs · 2026-04-30

**Decision:**

Accept (regular)

**Comment:**

This paper studies Byzantine-robust federated learning under partial participation, an important and still relatively underexplored setting. Reviewers generally agreed that the core idea of delayed momentum aggregation is simple, well motivated, and technically meaningful, and that the submission makes a useful contribution toward robust FL when sampled clients may contain a Byzantine majority. Overall, three of the four reviewers support acceptance, and one reviewer explicitly indicated that the rebuttal resolved their concerns, although they did not adjust the score.

The discussion focused mainly on the breadth of the experimental evaluation, the server-side memory cost, the lack of stronger adaptive-attack experiments, and the practical sharpness of some of the theoretical conditions. During rebuttal, the authors engaged seriously with these points: they clarified the algorithmic presentation, added the requested comparison with prior work in a fixed-size sampling regime where Byzantine-majority rounds do not occur, and responded in detail on memory cost, preprocessing, breakdown conditions, and multiple local updates. I agree that the paper still has limitations, especially regarding larger-scale validation and more targeted adaptive attacks, and these should be clearly discussed in the final version.

Importantly, after the initial round of final justifications, the authors posted an additional clarification prompted by reviewer pAun’s feedback. In that response they reported a configuration issue in the experimental sampling routine, corrected the affected results, and showed that the qualitative conclusions remain unchanged. They also strengthened the theory by removing the previous dependence on the absolute number of honest clients from the main result. Reviewer pAun’s remaining weak reject appears not to incorporate this last clarification, since the reviewer’s final justification predates it. I have asked explicitly reviewer pAun whether the new response changed their view, but they did not answer.

Overall, while the paper does not fully resolve every concern raised by Reviewer pAun, especially regarding adaptive attacks and broader empirical validation, I do not view these issues as fatal to the core contribution. The paper introduces a novel and potentially impactful idea for an important problem, the theoretical contribution was materially strengthened during discussion, and the corrected experiments preserve the main conclusions. On balance, I recommend acceptance.